# Improved Gravitational Search Algorithm Based on Adaptive Strategies

**DOI:** 10.3390/e24121826

**Published:** 2022-12-14

**Authors:** Zhonghua Yang, Yuanli Cai, Ge Li

**Affiliations:** 1College of Systems Engineering, National University of Defense Technology, Changsha 410073, China; 2Faculty of Electronics & Information, Xi’an Jiaotong University, Xi’an 710049, China

**Keywords:** gravitational search algorithm, swarm intelligence algorithm, adaptive strategy, particle information interaction

## Abstract

The gravitational search algorithm is a global optimization algorithm that has the advantages of a swarm intelligence algorithm. Compared with traditional algorithms, the performance in terms of global search and convergence is relatively good, but the solution is not always accurate, and the algorithm has difficulty jumping out of locally optimal solutions. In view of these shortcomings, an improved gravitational search algorithm based on an adaptive strategy is proposed. The algorithm uses the adaptive strategy to improve the updating methods for the distance between particles, gravitational constant, and position in the gravitational search model. This strengthens the information interaction between particles in the group and improves the exploration and exploitation capacity of the algorithm. In this paper, 13 classical single-peak and multi-peak test functions were selected for simulation performance tests, and the CEC2017 benchmark function was used for a comparison test. The test results show that the improved gravitational search algorithm can address the tendency of the original algorithm to fall into local extrema and significantly improve both the solution accuracy and the ability to find the globally optimal solution.

## 1. Introduction

With the progress of science and technology as well as the development of production and management, optimization problems cover almost all aspects of human life and production, becoming an important theoretical basis and indispensable method of modern science. The main solutions to optimization problems include traditional optimization methods and modern optimization methods. Traditional optimization methods are based on single-point optimization, and the main approaches are enumeration methods, numerical methods, and analytical methods. Modern optimization methods use swarm intelligence algorithms, inspired by the stimulation of biological evolution, that simulate the structural characteristics, evolutionary laws, thinking structures, and behavior patterns of human, natural, and other biological populations. Typical swarm intelligence algorithms include the evolutionary algorithm, artificial immune algorithm, memory algorithm, particle swarm optimization, shuffled frog leaping algorithm, cat swarm optimization, bacterial foraging optimization, artificial fish school algorithm, ant colony algorithm, and artificial bee colony algorithm. Traditional optimization methods have strict requirements for the optimization problems in practical projects, and the calculation speed is slow and the convergence is poor when solving large-scale complex problems. Often, the solution to the problem cannot be found in an acceptable time. Modern optimization methods have loose requirements for solving problems, and have good adaptability, robustness, and global search ability.

The gravitational search algorithm (GSA) was proposed by Raahedi et al. in 2009. It is a swarm intelligence global optimization algorithm that is simple to implement and achieves relatively good performance in global search and convergence. It is widely used in path planning [1,2], image classification [3,4,5], neural networks [6,7], data prediction [8,9,10], scheduling and parameter estimation [11,12,13,14,15,16,17], and other fields. However, the solution accuracy is not high, and the algorithm finds it difficult to jump out of locally optimal solutions in the later stages. To solve the shortcomings of this algorithm, scholars have improved the GSA algorithm with respect to the following three aspects: improvements to the adaptive strategy, integration with other swarm intelligence algorithms, and the introduction of other improvement strategies.

To address the first aspect, the authors of [18] proposed an adaptive GSA called SGSA in which an exponential decay model was introduced to the gravitation constant so that the algorithm can adjust the relevant parameters as required by the algorithm as the iteration proceeds. This also improves the exploration ability of the algorithm. An adaptive strategy based on population density was proposed for the distance between particles to prevent the algorithm from degenerating into random motion and to accelerate the convergence speed. The authors of [19] designed a new dynamic inertial weight and velocity position trend factor to improve the GSA, so that the inertial mass of particles has a certain trend as the iterations progress. This gives the change in position of each particle randomness and stability and gives the algorithm a certain degree of adaptability.

To address the second aspect, the authors of [20] combined the GSA with an immune algorithm, which introduces antibody diversity and immune memory characteristics into GSA and improves its global search ability. To overcome the problems of slow iterations and tendency to fall into local minima during the optimization of the standard GSA, one study [21] introduced the speed update mechanism of particle swarm optimization into the position update of GSA, combining the exploitation ability of particle swarm optimization and the exploration ability of GSA, and effectively solving the abovementioned problems. Another study [22] combined the free search differential evolution algorithm with the GSA to make full use of the exploration ability of GSA and the exploitation ability of the free search differential evolution algorithm, and to avoid the premature convergence of GSA. The authors of [23] combined the GSA with the sperm swarm optimization algorithm, which combines the advantages of both algorithms. Through testing, the hybrid method was found to have a better ability to avoid local extrema, and its convergence speed is relatively fast.

Finally, several studies have addressed the third aspect. The authors of [24] introduced a mutation operator to GSA and performed the mutation operation on particles with poor fitness values in the population. The particles were reinitialized, effectively preventing the algorithm from falling into locally optimal values, and the improved algorithm was successfully applied to an economic load scheduling problem. Another study [25] proposed the rotation GSA, which optimizes the selection of the k best in GSA by introducing a rotation operator so that unincluded particles have the opportunity to affect the motion of other particles and to enhance the exploration ability of the algorithm. The study [26] proposed a GSA based on Levy flight and chaos theory. The Levy distribution was used to improve the diversity of the population search space, and chaos search was used to strengthen candidate solutions to achieve global optimization. The authors of [27] proposed an improved GSA based on mutation strategy and reverse evaluation mechanism. The reverse learning bidirectional evaluation mechanism proposed by Tizhoosh was used to initialize and update the population so that particles were better distributed. In addition, the best individual and particles with poor fitness were cross-mutated using a mutation strategy to avoid premature convergence.

In summary, the key to improving the performance of the GSA is to balance the diversity and convergence of particles and prevent the algorithm from falling into local extreme values too early. Among the above three aspects of improvement, the adaptive strategy has a better effect and obtains the best performance. However, scholars have only used two adaptive strategies at most to improve the algorithm performance. Hence, there is room for further improvements in the algorithm’s performance. In this paper, a new adaptive GSA is proposed in which three adaptive strategies for population density, gravitation constant, and location update are used in combination to improve the optimization accuracy and convergence of the GSA. The organizational structure of this paper is as follows: Section 2 outlines the basic GSA, Section 3 introduces the three adaptive improvement strategies, Section 4 describes the idea and steps of the improved GSA and analyzes the space–time complexity and convergence performance. Finally, Section 5 evaluates the performance of the improved GSA through experiments, and Section 6 summarizes the conclusions.

## 2. Basic GSA

The universal GSA treats all particles as objects with mass. During the optimization process, all particles move unimpeded. Each particle is affected by the gravity of the other particles in the solution space and generates acceleration to move toward the particles with greater mass. Because the mass of the particles is related to their fitness, particles with large fitness will have greater mass. Therefore, particles with small masses will gradually approach the optimal solution in the optimization problem in the process of approaching particles with large masses. The GSA is different from other swarm intelligence algorithms. In the GSA, particles do not need to perceive the environment through environmental factors but realize information sharing through the interaction of the gravitational forces between individuals. Therefore, without the influence of environmental factors, particles can also perceive the global situation to conduct a global search in the environment, thus realizing the global optimization of the problem.

In a GSA, we assume that a D-dimensional search space contains *N* objects, and the position of the *i*-th object is
(1)$$X_{i}=(x_{i}^{1},x_{i}^{2},x_{i}^{3},...,x_{i}^{k}...x_{i}^{D}),i=1,2,...,N$$

In Equation (1), $x_{i}^{k}$ represents the position of the *i*-th object in the *k*-th dimension.

### 2.1. Inertial Mass Calculation

In the GSA, the inertial mass of each particle is directly related to the fitness value obtained from the particle location. At time *t*, the mass of particle $X_{i}$ is expressed by $M_{i}(t)$. Because the inertial mass *M* is calculated according to its corresponding fitness value, the particles with larger *M* values are closer to the optimal solution in the solution space, and they exert a greater attraction on other objects.

Particle mass $M_{i}(t)$ is calculated according to
(2)$$\left\{\begin{array}{c} m_{i}(t)=\frac{fit_{i}(t)-worst(t)}{best(t)-worst(t)}\\ M_{i}(t)=\frac{m_{i}(t)}{\sum_{j=1}^{N}m_{j}(t)} \end{array}\right.$$

Here, $fit_{i}(t)$ represents the fitness of particle $X_{i}$, best(t) represents the best solution at time *t*, worst(t) represents the worst solution at time *t*, and the calculation is as follows:(3)$$\left\{\begin{array}{c} best(t)=\underset{i\in \{1,2,...,N\}}{\max fit(t)}\\ worst(t)=\underset{i\in \{1,2,...,N\}}{\min fit(t)} \end{array}\right.$$

It can be seen from Equation (2) that $m_{i}(t)$ normalizes the fitness of particles to the range [0, 1] and then takes its proportion in the total mass as the mass $M_{i}(t)$ of the particles.

### 2.2. Gravitational Calculation

At time *t*, the calculation of the gravitational force of object *j* subjected to object *i* in the *k*-th dimension is as follows:(4)$$F_{ij}{}^{k}(t)=G(t)\frac{M_{ai}(t)\times M_{aj}(t)}{R_{ij}{}^{}(t)+\varepsilon }(x_{j}^{k}(t)-x_{i}^{k}(t)),$$
where ε represents a very small constant, *M_aj_*(*t*) represents the inertial mass of the action object *j*, and *M_ai_*(*t*) represents the inertial mass of the action object *i*. Furthermore, *G*(*t*) represents the constant of universal gravitation transformed over time, where its size is related to the number of iterations, and its calculation is
(5)$$G(t)=G_{0}\times e^{-\alpha t/T}.$$

In Equation (5), *G*_0_ represents the value of *G* at time *t*_0_, where *G*_0_ = 100, α = 20, and *T* is the maximum number of iterations. Finally, $R_{ij}(t)$ represents the Euclidean distance between objects $X_{i}$ and $X_{j}$, and is calculated as
(6)$$R_{ij}(t)=\left\|X_{i}\right.(t)-{\left.X_{j}(t)\right\|}_{2}.$$

At time *t*, the force acting on $X_{i}$ in the *k*-th dimension is equal to the sum of the forces exerted on it by all other particles around as follows:(7)Fki(t)=∑j=1,j≠iNrandFjijk(t).

### 2.3. Location Update

When a particle is subjected to the gravitational action of other particles, it will generate acceleration. According to the gravity calculated in Equation (7), the acceleration obtained by object *i* in the *k*-th dimension is the ratio of its force to inertial mass. The calculation is as follows:(8)$$\alpha_{i}^{k}(t)=\frac{F_{i}^{k}(t)}{M_{i}(t)}.$$

In each iteration, the algorithm updates the speed and position of object *i* according to the calculated acceleration. The update method is
(9)$$v_{i}^{k}(t+1)=rand_{i}\times v_{i}^{k}+\alpha_{i}^{k}(t),$$
(10)$$x_{i}^{k}(t+1)=x_{i}^{k}(t)+v_{i}^{k}(t+1).$$

The basic GSA implementation steps are as follows:Initialize the position and acceleration of all particles in the algorithm, and set the number of iterations and parameters.Calculate the fitness value for each particle, and update the gravitation constant according to the formula.The mass of each particle is calculated according to the calculated fitness value, and the acceleration of each particle is calculated using Equations (2)–(8).Calculate the speed of each particle according to Equation (9) and then update the particle position according to Equation (10).If the termination condition is not met, return to step 2; otherwise, output the optimal solution of the algorithm.

## 3. Adaptive Strategies

### 3.1. Adaptive Population Density Strategy

The distance between particles in the basic GSA is the Euclidean distance. Through a large number of experiments in [18], it was found that a constant fixed distance is better than the Euclidean distance, but the fixed distance value has obvious shortcomings: First, when the population is divided, the distance between particles is large and the interaction force is very small, and hence, the GSA degenerates into random motion. Second, when the population is dense, the distance between particles is very small, and the interaction force is very large. The particles in the algorithm will oscillate at a high frequency near the optimal solution and reduce the convergence speed. The population density is an indicator for evaluating the distance between particles, and it is the median of the average distance of all particles in the population. A smaller population density means the population is more concentrated; by contrast, a higher population density means the population is more dispersed. To solve the above two issues, balance the exploration and exploitation abilities of the algorithm, adjust the search ability of the algorithm, and propose an adaptive strategy based on population density, we dynamically adjust the distance between particles according to the GSA population density. That is, when the population density is relatively large, the population is relatively dispersed, reducing the particle distance between populations, promoting information exchange between particles, and preventing random movement between particles. When the population density value is small, the population is dense. We hence increase the distance between population particles appropriately to speed up the convergence of the algorithm. The calculation of population density δ is as follows:(11)$$\delta =\frac{1}{N}\sum_{i=1}^{N}dis_{i},$$
where *N* is the number of particles, *D* is the dimensionality of the particles, and *dis_i_* is the average distance between the *i*-th particle and all other particles, calculated as follows:(12)$${\mathrm{dis}}_{i}=\frac{1}{N-1}\sum_{j=1,j\neq i}^{N}\sqrt{\sum_{k=1}^{D}{\left(x_{i}^{k}-x_{j}^{k}\right)}^{2}}.$$

The gravitational force calculated in the basic universal GSA is modified as follows:(13)$$F_{ij}{}^{k}(t)=G(t)\frac{M_{ai}(t)\times M_{aj}(t)}{R_{ij}{}^{\mathrm{Rp}(\delta )}(t)+\varepsilon }(x_{j}^{k}(t)-x_{i}^{k}(t)).$$

The calculation of Rp(δ) is
(14)$$\mathrm{Rp}(\delta )=\left\{\begin{array}{c} Rp_{\min}+(Rp_{\max}-Rp_{\min})e^{1-1/\delta }\begin{array}{cc} & \delta <1 \end{array}\\ Rp_{\min}+(Rp_{\max}-Rp_{\min})e^{1-\delta }\begin{array}{cc} & \delta \geq 1 \end{array} \end{array}\right.,$$
where *Rp*_max_ and *Rp*_min_ are the maximum and minimum values of the given fixed distance, respectively, and δ is the population density.

### 3.2. Adaptive Gravitational Constant Strategy

Gravitational constant *G* is an important variable that transforms over time. Its change directly affects the magnitude of resultant force and acceleration, as well as determines the current step size and convergence speed of particles in the algorithm. The reasonable selection of parameters *G*_0_ and α plays an important role in the size of the iterative steps in the algorithm and determines whether the algorithm can jump out of local optima and determine the direct factor of optimal accuracy. If the original gravitational constant decreases quickly at the beginning of the algorithm, the algorithm can converge quickly, but it also tends to fall into the local optima and is difficult to jump out. To improve the exploration ability of the algorithm, prevent the algorithm from falling into locally optimal solutions, and improve the accuracy of the solution, an adaptive strategy for the universal gravitational constant is proposed. The adaptive gravitational constant *G* is expressed as follows:(15)$$G(t)=\frac{G_{0}^{}}{1{+e}^{\alpha (t-t_{c})/T}}0\leq t_{c}<T$$

Here, $G_{0}^{}$ is the initial value of the universal gravitational constant, α is the parameter of the decay rate, *T* is the total number of iterations, and *t_c_* is a constant value in the interval [0, *T*).

### 3.3. Adaptive Location Update Strategy

A position in the basic GSA is updated according to the current speed of the algorithm and the position in the last iteration. In each iteration, if the current update speed of particles is small, the change in position will also be small, the convergence ability of the algorithm will be reduced, and the algorithm will tend to fall into local extrema. By contrast, if the current update speed of particles is too large, the change in position will also increase, and the algorithm will move far from the global optimum. To address these defects, the improved strategy in [3] is adopted in this study to make the position of the particles change with respect to the iterative evolution. In the early stages of the algorithm, each particle moves with a large step size so that the algorithm can quickly converge to the vicinity of the optimal solution; in the later stages of the algorithm, the particle update step is smaller, and the particle depth search is near the optimal value. The expression of the adaptive position update is
(16)$$x_{i}^{k}(t+1)=\alpha \times x_{i}^{k}(t)+\beta \times v_{i}^{k}(t+1),$$
where α is calculated by
(17)$$\alpha =e^{\left(-\dim*{\left(t/T_{\max}\right)}^{\omega }\right)}$$
and β is calculated by
(18)$$\beta =1-\frac{t}{T_{\max}+betarnd},$$
where dim is the dimension; ω is an integer in the range [1, 50]; *T* is the current number of iterations of the algorithm; *T*_max_ is the maximum number of iterations set for the algorithm; and *betarnd* is the random number generated by the [0, 1] beta distribution. The range of α is (0, 1) and the range of β is (0, 1).

## 4. Improved GSA Based on Adaptive Strategies

### 4.1. Basic Concept

To improve the low solution accuracy and difficulty of jumping out of the locally optimal solutions of conventional GSA, the parameters for the distance between particles, gravitational constant, and position update in GSA are improved using adaptive strategies to strengthen the information interaction between particles and improve the exploration and exploitation capabilities of the algorithm.

The steps of the proposed algorithm are as follows.

**Step 1:** The proposed adaptive GSA is initialized to generate the initial particle swarm. Set the size of algorithm particle swarm *N* and the maximum number of iterations *NC*_Max_, search space dimension *X*_Dim_, maximum distance *Rp*_max_, minimum distance *Rp*_min_, gravitational constant, attenuation rate, constant value, and other parameter values.**Step 2:** Check the particle boundary in the population, and calculate the fitness value of particles in the population.**Step 3:** Use Equation (3) to calculate *best*(t) and *worst*(t).**Step 4:** The inertial mass *M_i_*(t) of the particles is obtained according to *best*(t), *worst*(t), and Equation (2).**Step 5:** Update the gravitational constant *G* according to Equation (15).**Step 6:** Calculate the distance between particles according to Equations (6), (11), and (14).**Step 7:** Calculate the gravitational and resultant forces around particles according to Equation (13).**Step 8:** Calculate the acceleration of the particles according to Equation (8).**Step 9:** Update particle speeds and positions according to Equations (9) and (16)–(18).**Step 10:** Return to the iteration cycle in step 2 until the number of cycles or accuracy requirements are met.**Step 11:** Exit the loop and output the algorithm results.

### 4.2. Temporal and Spatial Complexity Analyses

#### 4.2.1. Time Complexity Analysis

The time complexity of the algorithm is the time spent executing the algorithm, which is equal to the cumulative number of times the algorithm performs basic operations such as addition, subtraction, multiplication, division, and comparison. Assuming that the particle swarm size of the proposed adaptive GSA is *N*, the time complexity of the algorithm is analyzed according to the steps of the algorithm execution using the method in [28].

In step 1, the initialization of the particle swarm of the proposed adaptive GSA requires *N* operations, and the initialization operations of the other parameters are a constant; thus, the time complexity of step 1 is O (*N*).

In step 2, the particle boundary check requires *N* operations, the fitness calculation requires *N* operations, and hence, the time complexity of step 2 is O(*N*) + O(*N*).

In step 3, it takes one operation to calculate the best fitness value *best*(*t*) and one operation to calculate the worst fitness value *worst*(*t*), and hence, the time complexity of step 3 is O(1) + O(1).

Calculating the inertial mass *M_i_*(*t*) of particles in step 4 requires *N* operations; thus, the time complexity of step 4 is O(*N*).

Updating the gravitational constant *G* in step 5 requires one operation; thus, the time complexity of step 5 is O(1).

In step 6, calculating the average distance of all particles requires *N* × (*N* − 1) operations, calculating the population density requires one operation, and calculating the particle distance requires one operation, and hence, the time complexity of step 6 is O(*N* × (*N* − 1)) + O(1) + O(1).

In step 7, calculating the gravity of particles in the population requires *N* operations, and calculating the resultant force of particles in the population requires *N* operations; thus, the time complexity of step 7 is O(*N*) + O(*N*).

In step 8, calculating particle acceleration requires *N* operations, and calculating particle velocity requires *N* operations, and hence, the time complexity of step 8 is O(*N*) + O(*N*).

Updating particle positions in step 9 requires *N* operations; thus, the time complexity of step 9 is O(*N*).

In step 10, evaluating the termination condition requires one comparison operation, and terminating the algorithm requires one assignment operation; thus, the time complexity of step 10 is O(1) + O(1).

After the above steps, the proposed adaptive GSA performs *NC*_max_ iterations. The time complexity the proposed adaptive GSA after the maximum number of iterations is O(*NC*_max_ × (*N × N*)).

#### 4.2.2. Spatial Complexity Analysis

Space complexity is a measure of the storage space occupied by the algorithm during execution. Assume that the population size of the proposed adaptive GSA is *N*, the number of iterations of the algorithm is *NC*_max_, and the dimensionality of the optimization function is *D*. We perform spatial complexity analysis according to the steps of algorithm execution. In the proposed adaptive GSA, X [N] [D] is used to store the value of initialization independent variable, Y [1] [N] is used to store the fitness value of initialization function, Xm [1] [N] is used to store the inertial mass value of population particles, Xd [1] [N] is used to store the average distance between population particles, Xf [N] [D] is used to store the resultant force value around each particle in the population, Xa [N] [D] is used to store the acceleration value of each particle in the population, and Xv [N] [D] is used to store the velocity value of each particle in the population. Therefore, the space complexity of the whole GSA based on adaptive strategy improvement is 4 × O(N × D).

#### 4.2.3. Analysis of Algorithm Convergence

The convergence of the proposed adaptive GSA is proven using the contraction mapping theorem. For the relevant concepts and theorems in space compression theory, we refer readers to the definitions in [28].

 

**Theorem** **1****.**
*As the time tends to infinity, the proposed adaptive GSA is convergent.*


 

**Proof** **:**The state of the proposed adaptive GSA in the optimization process is represented by set *X*. The mutual transformation of states in set *X* is actually the embodiment of the whole optimization process of the proposed adaptive GSA. Therefore, the optimization process of the proposed adaptive GSA is a self-mapping process. If f is an optimization process mapping from *X* to *X*, then $X_{k+1}=f(X_{k})$. Suppose ∃ρ:X×X→R is the distance between two points in metric space (X,ρ) and $\left\{x_{n}\right\}$ is any optimization sequence in (X,ρ). □

 

The proposed adaptive GSA is a continuous iterative process. Under the action of gravity, individuals in the algorithm attract each other, forcing small mass individuals to constantly move to larger mass individuals to determine the optimal solution $X^{*}$. Therefore, in metric space (X,ρ), for any ε, there exists an N, where *n* > N, such that $\rho (x_{n},X^{*})<\varepsilon$ is true. According to its definition, the optimization sequence $\left\{x_{n}\right\}$ converges to $X^{*}$. Moreover, $\left\{x_{n}\right\}$ is a Cauchy sequence, and (X,ρ) is a complete metric space. Let ε be a random number in the range [0, 1]. Since the proposed adaptive GSA is a continuous optimization process, individuals are constantly approaching the optimal value, and hence, it is a convergent process. Then, there must be ρ(f(x),f(y)≤ε∗ρ(x,y) in the metric space (X,ρ), and f is a compression mapping. According to Theorem 4.2 in [28], $x^{*}=\underset{k\to \infty }{\lim}f^{*}(x_{0})$ is true, where $x^{*}\in X$ is the only fixed point in the compressed mapping f. Hence, the proposed adaptive GSA is convergent, and Theorem 1 is proven.

## 5. Experimental Analysis

### 5.1. Performance of the Algorithm Improvement Strategies

This section evaluates and analyzes the combinations of the three adaptive improvement strategies in the algorithm: the adaptive population density strategy, adaptive gravitational constant strategy, and adaptive location update strategy. For the convenience of description, we refer to the GSA based on the adaptive population density strategy as the RGSA. Similarly, we call the GSA based on the adaptive gravitational constant strategy the GGSA, and the GSA based on the adaptive position strategy is called the LGSA. The GSA based on the adaptive population density and gravity constant strategies is the RGGSA, the GSA based on the adaptive population density and location update strategies is the RLGSA, the GSA based on the adaptive gravity constant and position update strategies is the GLGSA, and the GSA based on all three adaptive population density, gravity constant, and location update strategies is the RGLGSA. In the experiment, we tested the convergence of the basic GSA and the GSA with the seven different adaptive strategies on benchmark functions f1 to f15.

#### 5.1.1. Test Functions

Table 1 lists the test functions. Among the 15 standard test functions, the minimum value 0.397887 is obtained at points (π, 12.275), (π, 2.275), and (9.42478, 2.475), and the other functions obtain the minimum value 0 at point (0, 0, ..., 0). The functions f1, f2, f3, f7, f8, f10, and f13 are unimodal test functions. These functions only have one globally optimal solution, and are mainly used to test the solution accuracy and development ability of the algorithm. Functions f4, f5, f6, f9, f11, f12, f14, and f15 are multimodal test functions. These functions have many local extrema. The GSA is prone to premature convergence or falling into local extrema. To determine the optimal values for these test functions, the algorithm must have the ability to jump out of local extrema, avoid premature convergence, and have strong global exploration ability.

#### 5.1.2. Data Analysis

We set the initial parameters of the algorithms as follows: for the basic GSA, G_0_ was 100 and α was 20; for RGSA: G_0_ was 100, $\alpha^{\text{·}}$ was 20, t_c_ was T/4, Rp_max_ was 1.5, and Rp_min_ was 0.5; for GGSA: G_0_ was 50 and $\alpha^{\text{·}}$ was 30; for SGSA: G_0_ was 100, α was 20, and ω was 10. The parameter settings of the other GSAs were consistent with those of the previous three GSAs. The population size for all algorithms was 50, the number of algorithm iterations was 1000, the test dimensions were 30 and 50, functions f6 and f11 were two-dimensional tests, and the number of independent algorithm runs was 30.

The following observations can be inferred from the simulation test results of Table 2, Table 3, Table 4, Table 5, Table 6, Table 7, Table 8, Table 9, Table 10, Table 11, Table 12, Table 13, Table 14, Table 15 and Table 16.

First, the results of the RGSA, GGSA, and LGSA on the test functions are better than those of the GSA. The results show that the three adaptive strategies of population density, gravitational constant, and location update can effectively improve the performance of the GSA. The detailed analysis is as follows. The result of the GGSA is inferior to those of the RGSA and LGSA, but superior to that of the GSA. This shows that although the adaptive gravitational constant strategy is inferior to the crowd density and location update strategy in improving the performance of the GSA, it also improves the performance of the GSA to a certain extent because it helps to improve the iteration step size and convergence speed. The result of the RGSA is much better than that of the GSA, which indicates that the adaptive population density strategy dynamically adjusts the distance between particles according to the population density in the evolution process and better balances the exploration and mining capabilities of the algorithm, thus improving the search capability of the algorithm. The LGSA is not only better than the GSA, but also better than the GGSA and RGSA, which shows that the location update strategy plays the largest role in improving the performance of the GSA and reflects that the balance of location and speed between individuals is the key to ensuring the good solution quality of a swarm intelligence algorithm.

Second, the results of the RGGSA, RLGSA, and GLGSA on the test function are better than those of the RGSA, GGSA, and LGSA. The results show that the combination of two of the three adaptive strategies proposed in this paper can effectively improve the performance of the GSAs using a single strategy. The result of the GLGSA is better than that of the RGGSA, but it is also inferior to that of the RLGSA, which indicates that the better single improvement strategy still has a higher performance advantage in the combined improvement strategy. The test result of the RGLGSA is higher than those of the RGGSA, RLGSA and GLGSA. The results shows that the RGSGSA, which combines the three strategies, leverages the advantages of the RGSA, GGSA, and LGSA, making the advantages more effective in the search process and achieving the best solution performance.

### 5.2. Comparison and Analysis of Algorithm Test Results

To fully evaluate the overall performance of the adaptive GSA proposed in this paper (RGLGSA), we selected the following classic and efficient GSA algorithms: the weight-based GSA (GSAGJ) [29], SGSA [18], and multipoint adaptive constraint-based gravitation improved algorithm (MACGSA) [19]. A comparative analysis of simulation tests was performed using 16 benchmark test functions of the CEC2017 benchmark. In the experiment, we tested the convergence of the basic GSA and the four comparison algorithms in 10, 30, and 50 dimensions.

#### 5.2.1. Test Function

In this study, 16 benchmark test functions of CEC2017 were selected to test the effectiveness of the proposed algorithm. Table 17 lists each test function. Among the 16 test functions, f19 and f23 take the minimum value 0 at point (1, 1, ..., 1), f29 and f30 take the minimum value 0 at point (−1, −1, ..., −1), and the other functions take the minimum value 0 at point (0, 0, ..., 0). Functions f16, f17, f18, f24, and f25 are multidimensional unimodal reference functions, whereas f19–f23 and f26–f31 are multidimensional multimodal reference functions.

#### 5.2.2. Data Analysis

To prevent errors caused by accidental factors and to ensure objectivity and fairness of the evaluation, in the experiment, the five algorithms were independently run 30 times and were iterated 1000 times in the same environment. The other parameter settings of the algorithms were consistent with those listed in Section 5.1.

The CEC2017 benchmark test function simulation results in Table 18, Table 19, Table 20, Table 21, Table 22, Table 23, Table 24, Table 25, Table 26, Table 27, Table 28, Table 29, Table 30, Table 31, Table 32 and Table 33 reveal the following.

First, for the unimodal functions f16, f18, f24, and f25, both the RGLGSA and MACGSA have high accuracy. Under the same number of dimensions, the optimization accuracy of the RGLGSA proposed in this paper is significantly higher than those of the GSA, GSAGJ, SGSA, and MACGSA. With different dimensions, as the dimensions increase, the solution accuracy of the five algorithms gradually decreases, but the solution accuracy of the RGLGSA is still higher than those of the GSA, GSAGJ, SGSA, and MACGSA. For function f17, both the RGLGSA and MACGSA have very high accuracy. Under the same number of dimensions, the optimization accuracy of the RGLGSA is superior to those of the other four algorithms. As the dimensions increase, the solution accuracy of the GSA, GSAGJ, and SGSA decrease gradually, with an obvious trend. The solution accuracies of the RGLGSA and MACGSA increase gradually, but the solution accuracy of the RGLGSA is still higher than that of the other four algorithms.

For the multimodal functions f19, f29, and f30, the four algorithms all become trapped in the local extrema, with little difference in solution accuracy. For functions f20, f21, f26, f27, and f28, the RGLGSA and MACGSA find the globally optimal solution 0, but other algorithms cannot. For function f22, when the dimensions are 10, the RGLGSA and MACGSA can find the global optimal solution 0; when the dimensions are high, only the RGLGSA can find the globally optimal solution 0. For function f23, when the dimensions are 10, the SGSA has a higher precision, and when the dimensions are 30 and 50, the RGLGSA has the highest precision. For function f31, under the same number of dimensions, the optimization accuracy of the RGLGSA is significantly higher than those of the other algorithms. As the dimensions increase, the solution accuracies of the GSA, GSAGJ, and SGSA decrease gradually. When the dimensions reach 50, the solution accuracies of the MACGSA and RGLGSA decrease significantly.

The comparison and test results of the above algorithms reveal that, compared with other classical and efficient improved GSAs, the RGLGSA has a relatively stable overall search ability on unimodal functions and multimodal functions. Moreover, it has a high convergence accuracy.

#### 5.2.3. Curve Analysis

Figure 1, Figure 2, Figure 3, Figure 4, Figure 5, Figure 6, Figure 7, Figure 8, Figure 9, Figure 10, Figure 11, Figure 12, Figure 13, Figure 14, Figure 15 and Figure 16 show the convergence curves of the RGLGSA, basic GSA, and the comparison GSAs. The convergence curves of the unimodal functions reveal that there is little difference between the convergence speed of the five algorithms in the first 500 generations. After 500 generations, the GSA, GSAGJ, and SGSA converge to the local extreme values and stop searching. The MACGSA and RGLGSA do not fall into the local extreme values, but continue to evolve. However, the convergence speed of the RGLGSA is faster than that of the MACGSA, and the obtained value is better.

For multimodal functions f19, f29 and f30, in the first 400 iterations, the convergence speed of the RLSGSA is roughly the same as those of the other four algorithms. Because of the characteristics of the functions and algorithms, after 400 iterations, the five algorithms converge to a local extremum, and the differences between the solutions obtained by the algorithms are not significant. For multimodal functions f20, f21, f27, and f28, GSA, GSAGJ, and SGSA converge to the local extreme value after a certain number of iterations and stop searching. The MACGSA and RGLGSA do not fall into the local extreme value but continue to evolve to find the globally optimal solution 0. However, the RGLGSA converges faster than the MACGSA. For multimodal function f22, in the first 500 iterations, the convergence speed of the RLSGSA is roughly the same as that of the other four algorithms. After 500 generations, the other four algorithms fall into local extrema, and the RLSGSA finds the global optimal solution 0 after about 650 generations. For multimodal function f31, in the first 300 iterations, the convergence rates of the five algorithms are similar. After 300 generations, the GSA, GSAGJ, and SGSA have stopped iterative evolutions, and the algorithms have fallen into locally optimal solutions. The MACGSA and RGLGSA continue to evolve after 600 generations, and finally, the MACGSA finds a better solution at a faster speed. For multimodal function f23, in 10 dimensions, the GSA and GSAGJ stop iterations after about 400 generations. After 400 generations, the convergence speed of the RGLGSA is faster than that of the SGSA and MACGSA. At 30 and 50 dimensions, the convergence accuracies of the five algorithms are not high. The GSAGJ has the fastest convergence speed, followed by the RGLGSA, but the convergence accuracy of the RGLGSA is higher than that of the GSAGJ. For multimodal function f26, the convergence speed of the MACGSA is higher before generation 500, the convergence speed of the RGLGSA is higher after generation 500, and the optimal value is obtained around generation 880. As a whole, the convergence speed of the RGLGSA is higher than those of the other comparison GSAs.

In general, compared with other GSA methods, the proposed adaptive GSA based on the adaptive strategies of population density, gravitational constant, and location update has a greatly improved optimization accuracy and stability, and performs well with respect to convergence.

## 6. Conclusions

In this paper, we proposed an improved GSA that is based on adaptive strategies. To address the shortcomings of the basic GSA, this algorithm introduces adaptive strategies based on crowd density, the gravitational constant, and location update into the basic universal GSA simultaneously. Moreover, it dynamically adjusts the distance between particles and the step size of the particle iteration, strengthens the information exchange between particles, and greatly increases the diversity of particles in the population. Therefore, it can effectively overcome the disadvantages of the basic GSA, which tends to fall into local extreme values. The simulation results show that the improved algorithm is superior to other classically improved GSAs in terms of search accuracy, convergence speed, stability, and other factors. It hence is an effective extension to the algorithm.

No Free Lunch Theory is a very important theorem in the field of optimization research, which reflects that no optimization algorithm can outperform other algorithms in average performance on all optimization problems. The improved algorithm in this paper also has this problem, and the experimental analysis cannot fully cover many optimization problems, which is convincing. The future research direction is mainly to optimize the overall performance of the algorithm, so that the performance of the improved algorithm is better than that of other algorithms on as many optimization problems as possible.

## Figures and Tables

**Figure 1 entropy-24-01826-f001:**
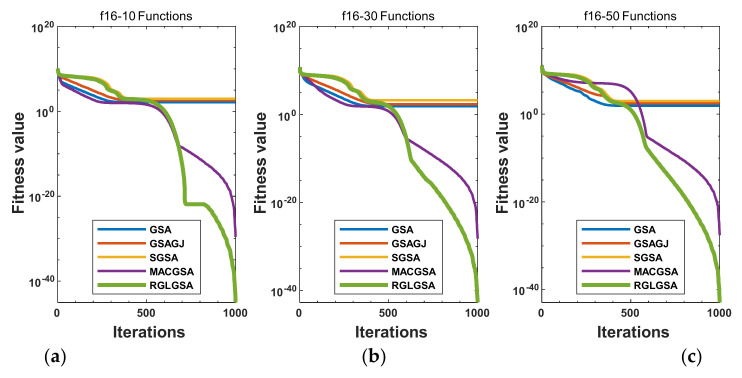
Convergence curves of function f16: (**a**) 10 dimensions, (**b**) 30 dimensions, and (**c**) 50 dimensions.

**Figure 2 entropy-24-01826-f002:**
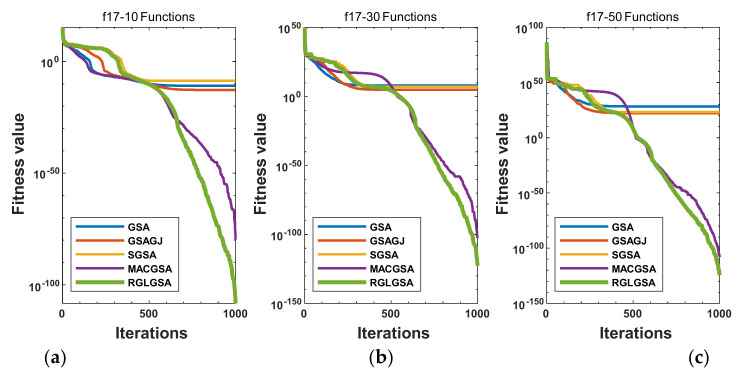
Convergence curves of function f17: (**a**) 10 dimensions, (**b**) 30 dimensions, and (**c**) 50 dimensions.

**Figure 3 entropy-24-01826-f003:**
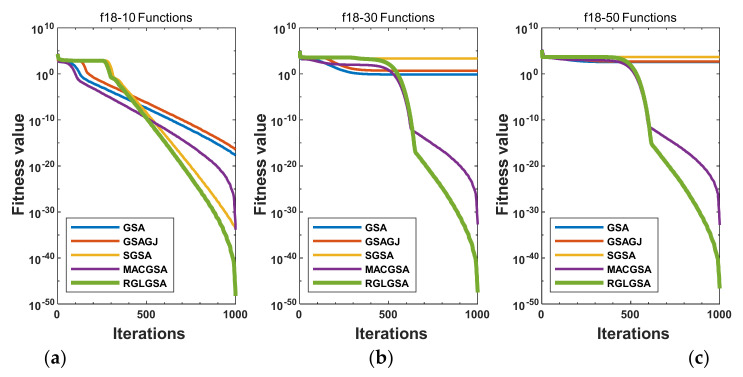
Convergence curves of function f18: (**a**) 10 dimensions, (**b**) 30 dimensions, and (**c**) 50 dimensions.

**Figure 4 entropy-24-01826-f004:**
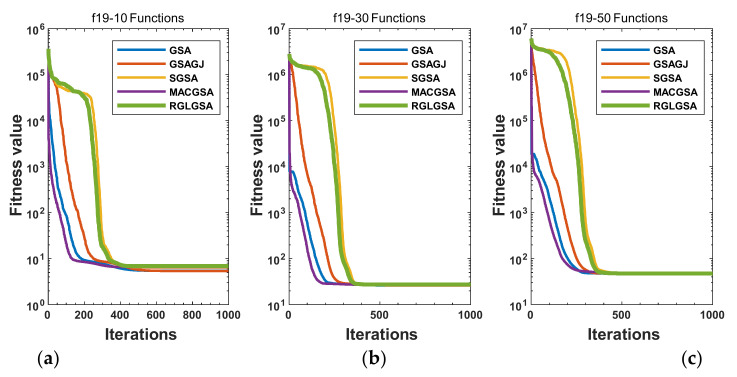
Convergence curves of function f19: (**a**) 10 dimensions, (**b**) 30 dimensions, and (**c**) 50 dimensions.

**Figure 5 entropy-24-01826-f005:**
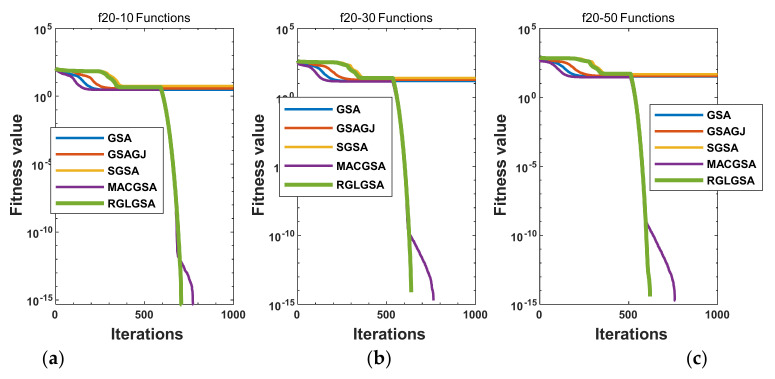
Convergence curves of function f20: (**a**) 10 dimensions, (**b**) 30 dimensions, and (**c**) 50 dimensions.

**Figure 6 entropy-24-01826-f006:**
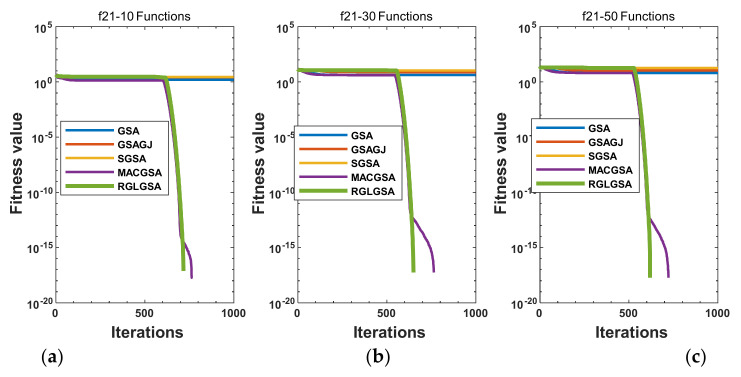
Convergence curves of function f21: (**a**) 10 dimensions, (**b**) 30 dimensions, and (**c**) 50 dimensions.

**Figure 7 entropy-24-01826-f007:**
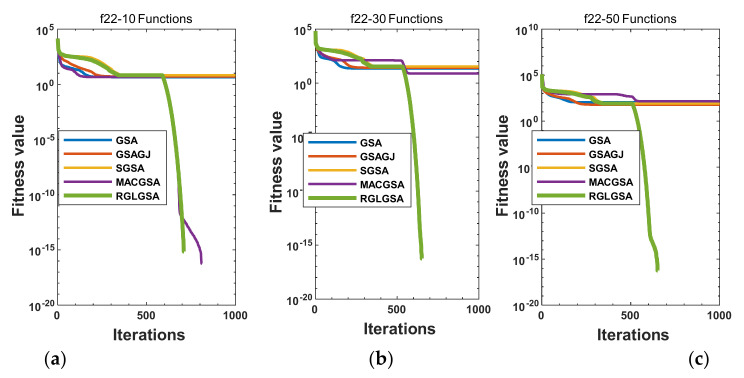
Convergence curves of function f22: (**a**) 10 dimensions, (**b**) 30 dimensions, and (**c**) 50 dimensions.

**Figure 8 entropy-24-01826-f008:**
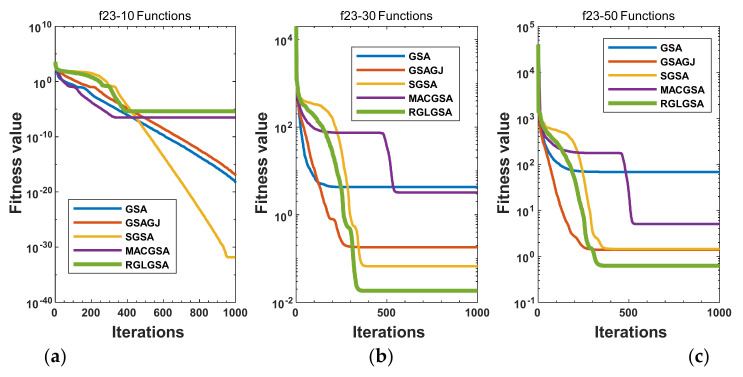
Convergence curves of function f23: (**a**) 10 dimensions, (**b**) 30 dimensions, and (**c**) 50 dimensions.

**Figure 9 entropy-24-01826-f009:**
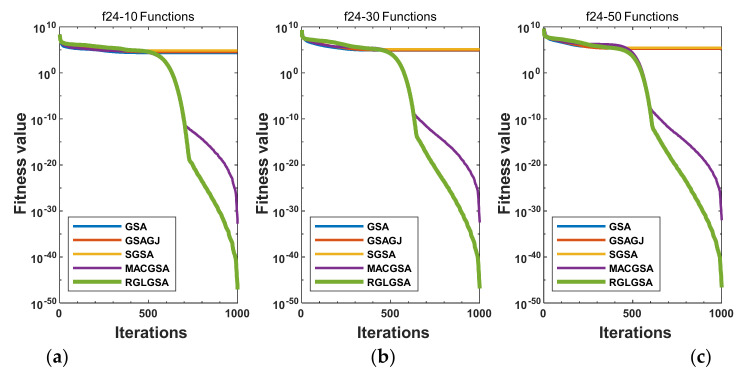
Convergence curves of function f24: (**a**) 10 dimensions, (**b**) 30 dimensions, and (**c**) 50 dimensions.

**Figure 10 entropy-24-01826-f010:**
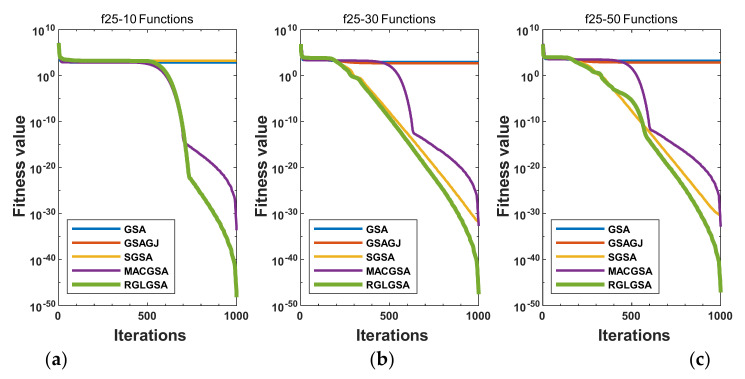
Convergence curves of function f25: (**a**) 10 dimensions, (**b**) 30 dimensions, and (**c**) 50 dimensions.

**Figure 11 entropy-24-01826-f011:**
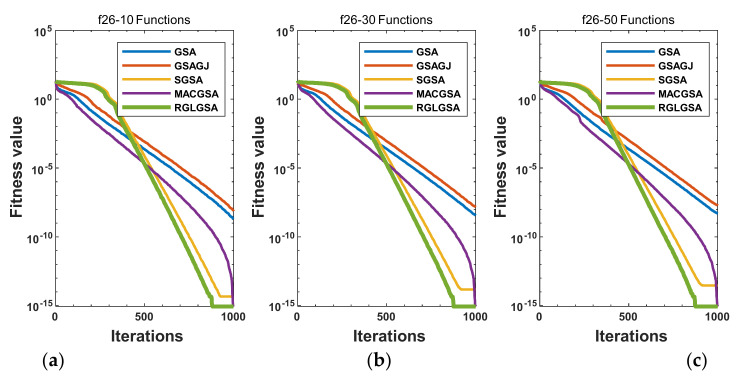
Convergence curves of function f26: (**a**) 10 dimensions, (**b**) 30 dimensions, and (**c**) 50 dimensions.

**Figure 12 entropy-24-01826-f012:**
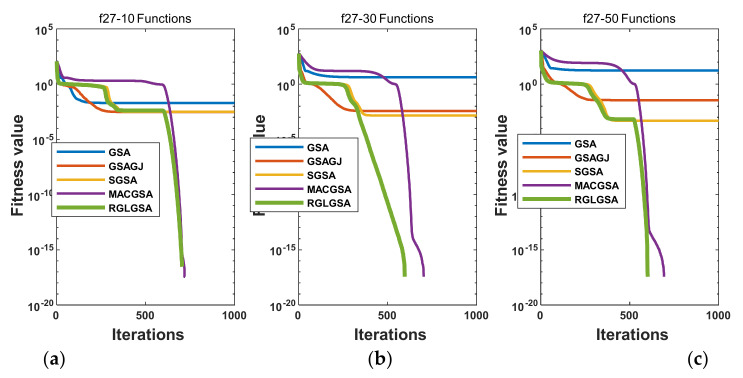
Convergence curves of function f27: (**a**) 10 dimensions, (**b**) 30 dimensions, and (**c**) 50 dimensions.

**Figure 13 entropy-24-01826-f013:**
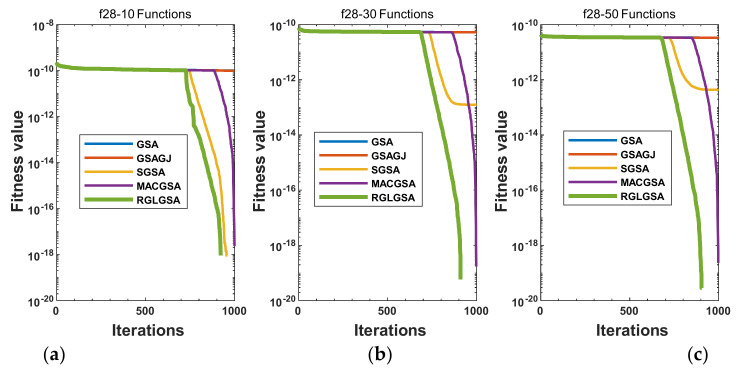
Convergence curves of function f28: (**a**) 10 dimensions, (**b**) 30 dimensions, and (**c**) 50 dimensions.

**Figure 14 entropy-24-01826-f014:**
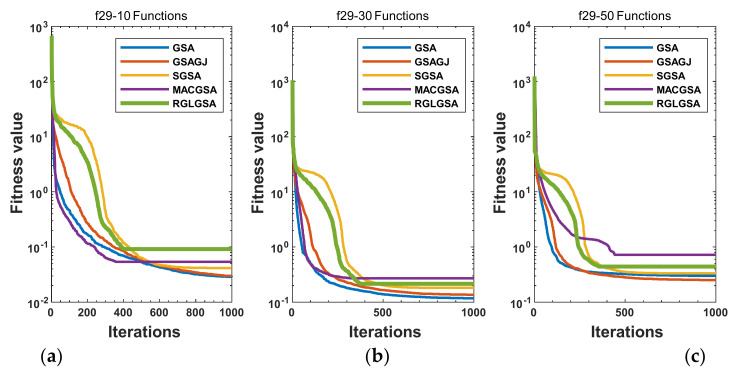
Convergence curves of function f29: (**a**) 10 dimensions, (**b**) 30 dimensions, and (**c**) 50 dimensions.

**Figure 15 entropy-24-01826-f015:**
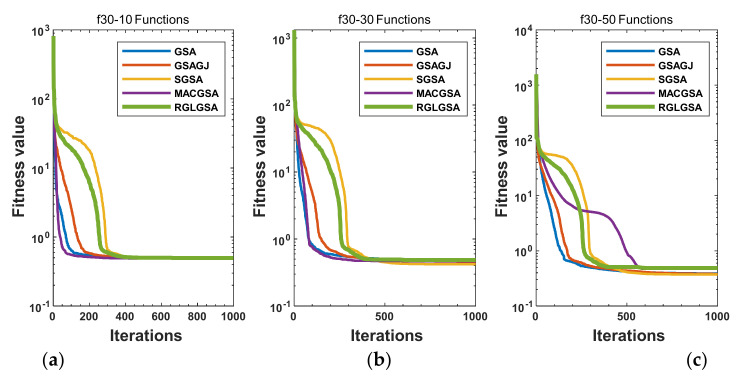
Convergence curves of function f30: (**a**) 10 dimensions, (**b**) 30 dimensions, and (**c**) 50 dimensions.

**Figure 16 entropy-24-01826-f016:**
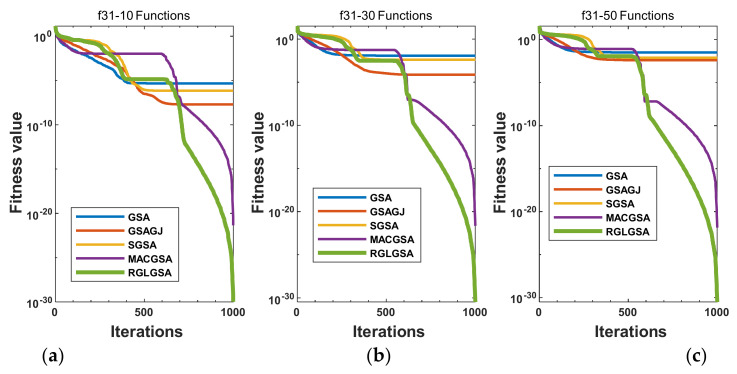
Convergence curves of function f31: (**a**) 10 dimensions, (**b**) 30 dimensions, and (**c**) 50 dimensions.

**Table 1 entropy-24-01826-t001:** Test function set.

Function No.	Test Function	Value Range
1	$f_{1}(x)=\sum_{i=1}^{n}x_{i}^{2}$	$x_{i}\in [-100,100]$
2	$f_{2}(x)=\sum_{i=1}^{n}\left|x_{i}\right|+\prod_{i=1}^{n}\left|x_{i}\right|$	$x_{i}\in [-10,10]$
	$f_{3}(x)=\max\left|x_{i}\right|$	$x_{i}\in [-100,100]$
4	$f_{4}(x)=-20\exp[-0.2\sqrt{\frac{1}{n}\sum_{i=1}^{n}x_{i}^{2}}]-\exp(\frac{1}{n}\sum_{i=1}^{n}\cos(2\pi x_{i}))+20+e$	$x_{i}\in [-32,32]$
5	$f_{5}(x)=\sum_{i=1}^{n}\left|\left.x_{i}\sin(x_{i})+0.1x_{i}\right|\right.$	$x_{i}\in [-10,10]$
6	$f_{6}(x)={\left(x_{2}-\frac{5.1}{4\pi }x_{1}^{2}-\frac{5}{\pi }x_{1}-6\right)}^{2}+10\times (1-\frac{1}{8\pi })\cos x_{1}+10$	$x_{i}\in [-10,10]$
7	$f_{7}(x)=\sum_{i=1}^{n}{\left(\sum_{j=1}^{i}x_{j}\right)}^{2}$	$x_{i}\in [-100,100]$
8	$f_{8}(x)=\sum_{i=1}^{n}ix_{i}^{4}+random\left[0,1\right]$	$x_{i}\in [-1.28,1.28]$
9	$f_{9}(x)=\frac{1}{4000}\sum_{i=1}^{n}x_{i}^{2}-\prod_{i=1}^{n}\cos(\frac{x_{i}}{\sqrt{i}})+1$	$x_{i}\in [-600,600]$
10	$f_{10}(x)=\sum_{i=1}^{n}x_{i}{}^{2}+{\left(\sum_{i=1}^{n}\frac{i}{2}x_{i}\right)}^{2}+{\left(\sum_{i=1}^{n}\frac{i}{2}x_{i}\right)}^{4}$	$x_{i}\in [-32,32]$
11	$f_{11}(x)=x^{2}+y^{2}+25(\sin^{2}x+\sin^{2}y)$	$x_{i}\in [-5.14,5.14]$
12	$f_{12}(x)=\sum_{i=1}^{n}\left[x_{i}^{2}-10\cos(2\pi x_{i})+10\right]$	$x_{i}\in [-5,5]$
13	$f_{13}(x)=\sum_{i=1}^{n}{\left(\left\lfloor x_{i}+0.5\right\rfloor \right)}^{2}$	$x_{i}\in [-100,100]$
14	$f_{14}(x)=\cos(2\pi \sqrt{\sum_{i=1}^{n}x_{i}^{2}})+0.1(\sqrt{\sum_{i=1}^{n}x_{i}^{2}})+1$	$x_{i}\in [-100,100]$
15	$f_{15}(x)=0.5+\frac{{\left(\sin\sqrt{x_{2}^{2}+x_{1}^{2}}\right)}^{2}-0.5}{{\left(1+0.001(x_{2}^{2}+x_{1}^{2})\right)}^{2}}$	$x_{i}\in [-10,10]$

**Table 2 entropy-24-01826-t002:** Test results for *f*_1_(*x*).

D	Algorithm	Worst	Best	Mean	SD
30	GSA	3.1784 × 10^−17^	1.1678 × 10^−17^	2.0655 × 10^−17^	4.9762 × 10^−18^
	RGSA	5.7262 × 10^−33^	4.1402 × 10^−34^	1.0540 × 10^−33^	1.0033 × 10^−33^
	GGSA	7.8267 × 10^−20^	1.6952 × 10^−20^	4.2861 × 10^−20^	1.5505 × 10^−20^
	LGSA	1.1171 × 10^−27^	7.0551 × 10^−31^	1.0968 × 10^−28^	2.3663 × 10^−28^
	RGGSA	2.4777 × 10^−38^	2.4197 × 10^−39^	8.1763 × 10^−39^	4.6307 × 10^−39^
	RLGSA	4.3833 × 10^−48^	4.9380 × 10^−51^	3.9451 × 10^−49^	9.0845 × 10^−49^
	GLGSA	2.2316 × 10^−29^	2.3715 × 10^−34^	9.0465 × 10^−31^	4.0589 × 10^−30^
	RGLGSA	**1.1227 × 10^−53^**	**1.3115 × 10^−56^**	**1.8360 × 10^−54^**	**3.3730 × 10^−54^**
50	GSA	1.0370 × 10^−16^	4.2637 × 10^−17^	7.1477 × 10^−17^	1.8306 × 10^−17^
	RGSA	7.9678 × 10^−32^	4.0968 × 10^−33^	1.7587 × 10^−32^	1.6047 × 10^−32^
	GGSA	3.1420 × 10^−19^	7.4754 × 10^−20^	2.0370 × 10^−19^	6.3958 × 10^−20^
	LGSA	7.8808 × 10^−28^	4.8414 × 10^−31^	8.3115 × 10^−29^	1.5517 × 10^−28^
	RGGSA	1.2945 × 10^−35^	1.7729 × 10^−37^	1.5203 × 10^−36^	2.7454 × 10^−36^
	RLGSA	3.2260 × 10^−48^	1.9879 × 10^−50^	6.0257 × 10^−49^	7.9558 × 10^−49^
	GLGSA	2.6738 × 10^−30^	2.0636 × 10^−33^	2.3400 × 10^−31^	5.0105 × 10^−31^
	RGLGSA	**1.4061 × 10^−53^**	**3.6091 × 10^−56^**	**2.1920 × 10^−54^**	**3.3828 × 10^−54^**

**Table 3 entropy-24-01826-t003:** Test results for *f*_2_(*x*).

D	Algorithm	Worst	Best	Mean	SD
30	GSA	2.9944 × 10^−8^	1.8259 × 10^−8^	2.3354 × 10^−8^	2.8498 × 10^−9^
	RGSA	3.7319 × 10^−16^	1.1320 × 10^−16^	2.2498 × 10^−16^	6.5816 × 10^−17^
	GGSA	1.5355 × 10^−9^	6.4974 × 10^−10^	1.0641 × 10^−9^	2.3299 × 10^−10^
	LGSA	5.3312 × 10^−14^	4.3968 × 10^−16^	1.2055 × 10^−14^	1.4035 × 10^−14^
	RGGSA	1.8821 × 10^−17^	3.8605 × 10^−19^	1.7182 × 10^−18^	3.3370 × 10^−18^
	RLGSA	5.1586 × 10^−24^	1.0922 × 10^−25^	8.5523 × 10^−25^	9.6857 × 10^−25^
	GLGSA	1.6789 × 10^−15^	1.0556 × 10^−16^	4.5455 × 10^−16^	3.1838 × 10^−16^
	RGLGSA	**4.8916 × 10^−27^**	**2.0799 × 10^−28^**	**1.8049 × 10^−27^**	**1.2726 × 10^−27^**
50	GSA	0.0236	3.7905 × 10^−08^	7.8561 × 10^−4^	0.0043
	RGSA	0.0861	6.2460 × 10^−16^	0.0047	0.0168
	GGSA	0.2069	2.1929 × 10^−9^	0.0181	0.0546
	LGSA	5.1145 × 10^−14^	1.2501 × 10^−15^	1.3527 × 10^−14^	1.1082 × 10^−14^
	RGGSA	0.2833	2.7805 × 10^−13^	0.0320	0.0748
	RLGSA	3.1275 × 10^−24^	1.8799 × 10^−25^	1.2276 × 10^−24^	8.5222 × 10^−25^
	GLGSA	2.8762 × 10^−15^	1.3163 × 10^−16^	6.1703 × 10^−16^	5.8851 × 10^−16^
	RGLGSA	**8.7338 × 10^−27^**	**1.9445 × 10^−28^**	**2.2760 × 10^−27^**	**1.8606 × 10^−27^**

**Table 4 entropy-24-01826-t004:** Test results for *f*_3_(*x*).

D	Algorithm	Worst	Best	Mean	SD
30	GSA	4.6203 × 10^−9^	2.0095 × 10^−9^	3.1109 × 10^−9^	7.1216 × 10^−10^
	RGSA	9.4873 × 10^−16^	5.6256 × 10^−17^	2.5335 × 10^−16^	2.4418 × 10^−16^
	GGSA	3.0792 × 10^−10^	9.8951 × 10^−11^	2.1632 × 10^−10^	4.9367 × 10^−11^
	LGSA	3.2172 × 10^−14^	6.3082 × 10^−16^	8.1911 × 10^−15^	8.6301 × 10^−15^
	RGGSA	0.0167	3.0764 × 10^−18^	0.0011	0.0041
	RLGSA	4.0438 × 10^−24^	2.3297 × 10^−26^	7.0978 × 10^−25^	8.7193 × 10^−25^
	GLGSA	2.9153 × 10^−15^	4.4903 × 10^−17^	5.0494 × 10^−16^	6.3364 × 10^−16^
	RGLGSA	**9.4406 × 10^−27^**	**2.0399 × 10^−28^**	**1.9853 × 10^−27^**	**1.9898 × 10^−27^**
50	GSA	7.3060	1.4952	3.8958	1.2656
	RGSA	7.7420	1.4171	3.8093	1.6933
	GGSA	0.0065	1.8832 × 10^−9^	2.4286 × 10^−4^	0.0012
	LGSA	2.9909 × 10^−14^	2.6038 × 10^−16^	9.7994 × 10^−15^	8.8800 × 10^−15^
	RGGSA	1.2411	0.3102	0.6938	0.2702
	RLGSA	6.4042 × 10^−24^	1.3099 × 10^−25^	1.2454 × 10^−24^	1.5236 × 10^−24^
	GLGSA	5.2383 × 10^−15^	2.7606 × 10^−17^	7.0144 × 10^−16^	1.0081 × 10^−15^
	RGLGSA	**1.0648 × 10^−26^**	**6.6210 × 10^−29^**	**3.0313 × 10^−27^**	**3.1224 × 10^−27^**

**Table 5 entropy-24-01826-t005:** Test results for *f*_4_(*x*).

D	Algorithm	Worst	Best	Mean	SD
30	GSA	4.5997 × 10^−9^	2.5140 × 10^−9^	3.5884 × 10^−9^	4.2747 × 10^−10^
	RGSA	1.5099 × 10^−14^	7.9936 × 10^−15^	1.0599 × 10^−14^	2.7886 × 10^−15^
	GGSA	2.5246 × 10^−10^	1.2959 × 10^−10^	1.6997 × 10^−10^	2.7635 × 10^−11^
	LGSA	9.3259 × 10^−14^	**8.8818 × 10^−16^**	1.1428 × 10^−14^	1.7165 × 10^−14^
	RGGSA	1.5099 × 10^−14^	7.9936 × 10^−15^	9.6515 × 10^−15^	2.5945 × 10^−15^
	RLGSA	**8.8818 × 10^−16^**	**8.8818 × 10^−16^**	**8.8818 × 10^−16^**	**0**
	GLGSA	4.4409 × 10^−15^	**8.8818 × 10^−16^**	1.1250 × 10^−15^	9.0135 × 10^−16^
	RGLGSA	**8.8818 × 10^−16^**	**8.8818 × 10^−16^**	**8.8818 × 10^−16^**	**0**
50	GSA	6.2569 × 10^−9^	3.7704 × 10^−9^	4.9280 × 10^−9^	6.2123 × 10^−10^
	RGSA	2.5757 × 10^−14^	1.5099 × 10^−14^	2.0191 × 10^−14^	3.1890 × 10^−15^
	GGSA	6.4020 × 10^−10^	1.9065 × 10^−10^	2.6571 × 10^−10^	8.2614 × 10^−11^
	LGSA	2.2204 × 10^−14^	**8.8818 × 10^−16^**	5.7436 × 10^−15^	4.4239 × 10^−15^
	RGGSA	0.0193	1.1546 × 10^−14^	0.0014	0.0043
	RLGSA	**8.8818 × 10^−16^**	**8.8818 × 10^−16^**	**8.8818 × 10^−16^**	**0**
	GLGSA	4.4409 × 10^−15^	**8.8818 × 10^−16^**	1.2434 × 10^−15^	1.0840 × 10^−15^
	RGLGSA	**8.8818 × 10^−16^**	**8.8818 × 10^−16^**	**8.8818 × 10^−16^**	**0**

**Table 6 entropy-24-01826-t006:** Test results for *f*_5_(*x*).

D	Algorithm	Worst	Best	Mean	SD
30	GSA	2.8447 × 10^−9^	1.3648 × 10^−9^	2.2171 × 10^−9^	3.7633 × 10^−10^
	RGSA	0.0045	3.8202 × 10^−18^	8.5381 × 10^−4^	0.0013
	GGSA	1.5427 × 10^−10^	7.2127 × 10^−11^	1.1032 × 10^−10^	1.8751 × 10^−11^
	LGSA	1.0398 × 10^−14^	7.2806 × 10^−17^	1.3223 × 10^−15^	2.1370 × 10^−15^
	RGGSA	0.0069	4.7484 × 10^−21^	0.0014	0.0021
	RLGSA	6.9037 × 10^−25^	2.8753 × 10^−26^	1.2272 × 10^−25^	1.3593 × 10^−25^
	GLGSA	2.3019 × 10^−16^	7.6262 × 10^−18^	6.1309 × 10^−17^	5.3079 × 10^−17^
	RGLGSA	**2.1912 × 10^−27^**	**3.3527 × 10^−29^**	**2.2977 × 10^−28^**	**4.0053 × 10^−28^**
50	GSA	0.0075	4.2822 × 10^−9^	7.8908 × 10^−4^	0.0017
	RGSA	0.0239	2.7429 × 10^−14^	0.0055	0.0053
	GGSA	0.0039	1.9820 × 10^−10^	7.3396 × 10^−4^	0.0011
	LGSA	8.0741 × 10^−15^	6.6396 × 10^−17^	1.2979 × 10^−15^	1.5773 × 10^−15^
	RGGSA	0.0361	1.4921 × 10^−5^	0.0105	0.0086
	RLGSA	6.5537 × 10^−25^	1.5424 × 10^−26^	1.4210 × 10^−25^	1.2736 × 10^−25^
	GLGSA	2.4325 × 10^−16^	1.1630 × 10^−17^	5.7302 × 10^−17^	4.5711 × 10^−17^
	RGLGSA	**9.6359 × 10^−28^**	**6.0262 × 10^−29^**	**3.0080 × 10^−28^**	**2.1554 × 10^−28^**

**Table 7 entropy-24-01826-t007:** Test results for *f*_6_(*x*).

D	Algorithm	Worst	Best	Mean	SD
2	GSA	**0.3979**	**0.3979**	**0.3979**	**0**
	RGSA	**0.3979**	**0.3979**	**0.3979**	**0**
	GGSA	**0.3979**	**0.3979**	**0.3979**	**0**
	LGSA	**0.3979**	**0.3979**	**0.3979**	**0**
	RGGSA	**0.3979**	**0.3979**	**0.3979**	**0**
	RLGSA	**0.3979**	**0.3979**	**0.3979**	**0**
	GLGSA	**0.3979**	**0.3979**	**0.3979**	**0**
	RGLGSA	**0.3979**	**0.3979**	**0.3979**	**0**

**Table 8 entropy-24-01826-t008:** Test results for *f*_7_(*x*).

D	Algorithm	Worst	Best	Mean	SD
30	GSA	441.6230	100.6353	232.0828	73.4886
	RGSA	3.0106 × 10^3^	905.0757	1.7284 × 10^3^	568.6537
	GGSA	173.7834	36.2260	101.6022	39.9589
	LGSA	2.2037 × 10^−24^	2.6581 × 10^−30^	7.9806 × 10^−26^	4.0151 × 10^−25^
	RGGSA	4.3546 × 10^3^	839.7732	2.3577 × 10^3^	895.2346
	RLGSA	4.2249 × 10^−46^	2.3761 × 10^−50^	7.2154 × 10^−47^	1.1422 × 10^−46^
	GLGSA	2.5695 × 10^−28^	5.9115 × 10^−33^	3.4041 × 10^−29^	7.3331 × 10^−29^
	RGLGSA	**1.0902 × 10^−51^**	**8.8591 × 10^−55^**	**2.3283 × 10^−52^**	**3.4958 × 10^−52^**
50	GSA	1.5986 × 10^3^	678.5357	988.3067	261.3114
	RGSA	8.8062 × 10^3^	3.8307 × 10^3^	5.6479 × 10^3^	1.2912 × 10^3^
	GGSA	865.9041	351.5170	642.5225	125.1997
	LGSA	1.0505 × 10^−24^	1.2817 × 10^−29^	7.9992 × 10^−26^	2.1493 × 10^−25^
	RGGSA	1.0125 × 10^4^	4.1745 × 10^3^	6.9878 × 10^3^	1.2641 × 103
	RLGSA	7.9867 × 10^−46^	1.2999 × 10^−49^	1.1830 × 10^−46^	1.8003 × 10^−46^
	GLGSA	8.3184 × 10^−28^	3.3753 × 10^−32^	9.8107 × 10^−29^	1.9489 × 10^−28^
	RGLGSA	**3.9046 × 10^−51^**	**9.6762 × 10^−56^**	**5.4271 × 10^−52^**	**8.8234 × 10^−52^**

**Table 9 entropy-24-01826-t009:** Test results for *f*_8_(*x*).

D	Algorithm	Worst	Best	Mean	SD
30	GSA	0.0386	0.0090	0.0193	0.0072
	RGSA	0.0587	0.0093	0.0301	0.0118
	GGSA	0.0547	0.0167	0.0310	0.0105
	LGSA	1.9174 × 10^−4^	8.5337 × 10^−8^	4.5438 × 10^−5^	4.6017 × 10^−5^
	RGGSA	0.0772	0.0161	0.0455	0.0156
	RLGSA	**1.1234 × 10^−4^**	7.6825 × 10^−7^	**3.7319 × 10^−5^**	**3.0182 × 10^−5^**
	GLGSA	2.0404 × 10^−4^	**1.1790 × 10^−7^**	5.0362 × 10^−5^	5.3880 × 10^−5^
	RGLGSA	1.8700 × 10^−4^	1.1549 × 10^−6^	4.5049 × 10^−5^	4.7092 × 10^−5^
50	GSA	0.1932	0.0296	0.0627	0.0317
	RGSA	0.2004	0.0517	0.1121	0.0359
	GGSA	0.1453	0.0307	0.0783	0.0290
	LGSA	2.9610 × 10^−4^	3.1457 × 10^−6^	5.6102 × 10^−5^	5.8344 × 10^−5^
	RGGSA	0.2718	0.0840	0.1449	0.0453
	RLGSA	1.8462 × 10^−4^	**6.1063 × 10^−7^**	3.6584 × 10^−5^	3.8892 × 10^−5^
	GLGSA	**1.1842 × 10^−4^**	2.1433 × 10^−6^	**3.3349 × 10^−5^**	**2.7926 × 10^−5^**
	RGLGSA	1.9608 × 10^−4^	7.4748 × 10^−7^	3.9030 × 10^−5^	4.4529 × 10^−5^

**Table 10 entropy-24-01826-t010:** Test results for *f*_9_(*x*).

D	Algorithm	Worst	Best	Mean	SD
30	GSA	7.4411	1.3746	4.0420	1.5575
	RGSA	0.0074	**0**	2.4653 × 10^−4^	0.0014
	GGSA	0.0537	**0**	0.0079	0.0162
	LGSA	**0**	**0**	**0**	**0**
	RGGSA	0.0123	**0**	9.8565 × 10^−4^	0.0031
	RLGSA	**0**	**0**	**0**	**0**
	GLGSA	**0**	**0**	**0**	**0**
	RGLGSA	**0**	**0**	**0**	**0**
50	GSA	23.4237	11.0850	17.2132	3.4714
	RGSA	0.0124	**0**	0.0015	0.0036
	GGSA	1.1759	**0**	0.1319	0.2313
	LGSA	**0**	**0**	**0**	**0**
	RGGSA	0.0099	**0**	3.5078 × 10^−4^	0.0018
	RLGSA	**0**	**0**	**0**	**0**
	GLGSA	**0**	**0**	**0**	**0**
	RGLGSA	**0**	**0**	**0**	**0**

**Table 11 entropy-24-01826-t011:** Test results for *f*_10_(*x*).

D	Algorithm	Worst	Best	Mean	SD
30	GSA	22.3692	9.7247	15.4151	2.8357
	RGSA	30.9017	10.7804	20.9584	5.1313
	GGSA	20.0970	6.6160	12.5334	3.7834
	LGSA	7.1106 × 10^−24^	7.4616 × 10^−30^	1.2373 × 10^−24^	1.7594 × 10^−24^
	RGGSA	45.7283	23.1280	35.3209	6.6800
	RLGSA	1.0930 × 10^−45^	2.7914 × 10^−47^	2.5198 × 10^−46^	2.5438 × 10^−46^
	GLGSA	1.3735 × 10^−26^	3.1561 × 10^−31^	2.2552 × 10^−27^	3.0441 × 10^−27^
	RGLGSA	**2.5063 × 10^−51^**	**7.9170 × 10^−53^**	**6.7030 × 10^−52^**	**6.1941 × 10^−52^**
50	GSA	49.1786	18.8750	31.4671	6.6325
	RGSA	47.8996	27.5320	37.6831	5.7390
	GGSA	42.9315	14.3102	29.6947	8.4815
	LGSA	2.1156 × 10^−23^	9.4851 × 10^−29^	4.9534 × 10^−24^	5.0170 × 10^−24^
	RGGSA	83.9796	55.0682	67.5294	8.7154
	RLGSA	2.2174 × 10^−45^	5.9626 × 10^−47^	5.5519 × 10^−46^	4.4312 × 10^−46^
	GLGSA	3.4122 × 10^−26^	1.9063 × 10^−30^	8.4759 × 10^−27^	9.3119 × 10^−27^
	RGLGSA	**6.6221 × 10^−51^**	**1.3981 × 10^−52^**	**1.5619 × 10^−51^**	**1.3807 × 10^−51^**

**Table 12 entropy-24-01826-t012:** Test results for *f*_11_(*x*).

D	Algorithm	Worst	Best	Mean	SD
2	GSA	6.9837 × 10^−19^	2.4931 × 10^−21^	1.4808 × 10^−19^	1.5281 × 10^−19^
	RGSA	1.5693 × 10^−36^	3.9431 × 10^−39^	3.6757 × 10^−37^	4.0331 × 10^−37^
	GGSA	1.1021 × 10^−21^	3.8066 × 10^−24^	2.6538 × 10^−22^	2.4632 × 10^−22^
	LGSA	2.5328 × 10^−25^	1.0902 × 10^−35^	2.4752 × 10^−26^	6.4866 × 10^−26^
	RGGSA	1.6362 × 10^−41^	4.2936 × 10^−44^	2.4011 × 10^−42^	3.8616 × 10^−42^
	RLGSA	3.1987 × 10^−47^	2.1284 × 10^−53^	1.9892 × 10^−48^	5.9666 × 10^−48^
	GLGSA	2.5843 × 10^−27^	3.2845 × 10^−35^	2.1927 × 10^−28^	5.3440 × 10^−28^
	RGLGSA	**4.3268 × 10^−53^**	**6.2023 × 10^−60^**	**5.7435 × 10^−54^**	**1.1301 × 10^−53^**

**Table 13 entropy-24-01826-t013:** Test results for *f*_12_(*x*).

D	Algorithm	Worst	Best	Mean	SD
30	GSA	21.8891	7.9597	14.8912	3.5586
	RGSA	34.8235	13.9294	22.2539	5.7095
	GGSA	26.8639	9.9496	18.5726	4.2961
	LGSA	**0**	**0**	**0**	**0**
	RGGSA	34.8235	14.9244	25.6699	5.8915
	RLGSA	**0**	**0**	**0**	**0**
	GLGSA	**0**	**0**	**0**	**0**
	RGLGSA	**0**	**0**	**0**	**0**
50	GSA	50.7429	22.8841	33.0326	6.0459
	RGSA	50.7429	23.8790	37.7089	7.0031
	GGSA	47.7580	21.8891	37.0125	6.4396
	LGSA	**0**	**0**	**0**	**0**
	RGGSA	81.5865	34.8235	51.2403	11.3435
	RLGSA	**0**	**0**	**0**	**0**
	GLGSA	**0**	**0**	**0**	**0**
	RGLGSA	**0**	**0**	**0**	**0**

**Table 14 entropy-24-01826-t014:** Test results for *f*_13_(*x*).

D	Algorithm	Worst	Best	Mean	SD
30	GSA	**0**	**0**	**0**	**0**
	RGSA	**0**	**0**	**0**	**0**
	GGSA	**0**	**0**	**0**	**0**
	LGSA	**0**	**0**	**0**	**0**
	RGGSA	**0**	**0**	**0**	**0**
	RLGSA	**0**	**0**	**0**	**0**
	GLGSA	**0**	**0**	**0**	**0**
	RGLGSA	**0**	**0**	**0**	**0**
50	GSA	4	**0**	0.6333	0.9994
	RGSA	5	**0**	0.6667	1.2130
	GGSA	**0**	**0**	**0**	**0**
	LGSA	**0**	**0**	**0**	**0**
	RGGSA	4	**0**	0.7667	1.0400
	RLGSA	**0**	**0**	**0**	**0**
	GLGSA	**0**	**0**	**0**	**0**
	RGLGSA	**0**	**0**	**0**	**0**

**Table 15 entropy-24-01826-t015:** Test results for *f*_14_(*x*).

D	Algorithm	Worst	Best	Mean	SD
30	GSA	2.4001	0.8014	1.2938	0.4433
	RGSA	1.2989	0.7001	0.9479	0.1567
	GGSA	0.9031	0.5999	0.6820	0.0680
	LGSA	5.6621 × 10^−15^	**0**	1.1361 × 10^−15^	1.5394 × 10^−15^
	RGGSA	1.4072	0.8052	1.1264	0.1450
	RLGSA	**0**	**0**	**0**	**0**
	GLGSA	1.1102 × 10^−16^	**0**	1.1102 × 10^−17^	3.3876 × 10^−17^
	RGLGSA	**0**	**0**	**0**	**0**
50	GSA	4.8987	2.4057	3.3326	0.5198
	RGSA	2.3001	1.4005	1.7868	0.2488
	GGSA	2.6005	0.9999	1.6749	0.3434
	LGSA	8.3267 × 10^−15^	1.1102 × 10^−16^	1.3582 × 10^−15^	1.7851 × 10^−15^
	RGGSA	2.5008	1.5183	1.8890	0.2373
	RLGSA	**0**	**0**	**0**	**0**
	GLGSA	2.2204 × 10^−16^	**0**	3.3307 × 10^−17^	5.9395 × 10^−17^
	RGLGSA	**0**	**0**	**0**	**0**

**Table 16 entropy-24-01826-t016:** Test results for *f*_15_(*x*).

D	Algorithm	Worst	Best	Mean	SD
2	GSA	0.0098	1.3534 × 10^−5^	0.0063	0.0039
	RGSA	0.0097	1.0306 × 10^−4^	0.0054	0.0038
	GGSA	0.0097	2.7770 × 10^−4^	0.0041	0.0032
	LGSA	**0**	**0**	**0**	**0**
	RGGSA	0.0097	5.8631 × 10^−4^	0.0057	0.0033
	RLGSA	**0**	**0**	**0**	**0**
	GLGSA	**0**	**0**	**0**	**0**
	RGLGSA	**0**	**0**	**0**	**0**

**Table 17 entropy-24-01826-t017:** CEC2017 benchmark function set.

Function No.	Test Function	Value Range
16	$f_{16}(x)=x_{1}^{2}+10^{6}\sum_{i=2}^{n}x_{i}^{2}$	$x_{i}\in [-100,100]$
17	$f_{17}(x)=\sum_{i=1}^{n}{\left|x_{i}\right|}^{i+1}$	$x_{i}\in [-100,100]$
18	$f_{18}(x)=\sum_{i=1}^{n}x_{i}^{2}+{\left(\sum_{i=1}^{n}0.5x_{i}\right)}^{2}+{\left(\sum_{i=1}^{n}0.5x_{i}\right)}^{4}$	$x_{i}\in [-100,100]$
19	$f_{19}(x)=\sum_{i=1}^{n-1}\left(100{\left(x_{i}^{2}-x_{i+1}\right)}^{2}+{\left(x_{i}-1\right)}^{2}\right)$	$x_{i}\in [-10,10]$
20	$f_{20}(x)=\sum_{i=1}^{n}\left(x_{i}^{2}-10\cos(2\pi x_{i})+10\right)$	$x_{i}\in [-5.12,5.12]$
21	$\begin{array}{l} g(x,y)=0.5+\frac{\left(\sin^{2}(\sqrt{x^{2}+y^{2}})-0.5\right)}{{\left(1+0.001(x^{2}+y^{2})\right)}^{2}}\\ f_{21}(x)=g(x_{1},x_{2})+g(x_{2},x_{3})+...+g(x_{n-1},x_{n})+g(x_{n},x_{1}) \end{array}$	$x_{i}\in [-100,100]$
22	$\begin{array}{l} y_{i}=\left\{\begin{array}{l} x_{i}\to \left|x_{i}\right|\langle \frac{1}{2}\\ round(2x_{i})/2\to \left|x_{i}\right|\geq \frac{1}{2} \end{array}\right.\\ f_{22}(x)=\sum_{i=1}^{n}\left(y_{i}^{2}-10\cos(2\pi y_{i})+10\right) \end{array}$	$x_{i}\in [-100,100]$
23	$\begin{array}{l} y_{i}=1+\frac{x_{i}-1}{4},\forall i=1,...,n\\ f_{23}(x)=\sin^{2}(\pi y_{1})+\sum_{i=1}^{n-1}{\left(y_{i}-1\right)}^{2}[1+10\sin^{2}(\pi y_{i}+1)]+{\left(y_{n}-1\right)}^{2}[1+\sin^{2}(2\pi y_{n})] \end{array}$	$x_{i}\in [-100,100]$
24	$f_{24}(x)=\sum_{i=1}^{n}{\left(10^{6}\right)}^{\frac{i-1}{n-1}}x_{i}^{2}$	$x_{i}\in [-100,100]$
25	$f_{25}(x)=\sum_{i=2}^{n}x_{i}{}^{2}+10^{6}x_{1}^{2}$	$x_{i}\in [-100,100]$
26	$f_{26}(x)=-20\exp(-0.2\sqrt{\frac{1}{n}\sum_{i=1}^{n}x_{i}^{2}})-\exp(\frac{1}{n}\sum_{i=1}^{n}\cos(2\pi x_{i}))+20+e$	$x_{i}\in [-32,32]$
27	$f_{27}(x)=\frac{1}{4000}\sum_{i=1}^{n}x_{i}^{2}-\prod_{i=1}^{n}\cos(\frac{x_{i}}{\sqrt{i}})+1$	$x_{i}\in [-600,600]$
28	$f_{28}(x)=\frac{10}{n^{2}}\prod_{i=1}^{n}(1+i\sum_{j=1}^{32}\frac{\left|2^{j}x_{i}-round(2^{j}x_{i})\right|}{2^{j}}{)}^{\frac{10}{n^{1.2}}}-\frac{10}{n^{2}}$	$x_{i}\in [-100,100]$
29	$f_{29}(x)={\left|\sum_{i=1}^{n}x_{i}^{2}-n\right|}^{1/4}+\left(0.5\sum_{i=1}^{n}x_{i}^{2}+\sum_{i=1}^{n}x_{i}\right)/n+0.5$	$x_{i}\in [-100,100]$
30	$f_{30}(x)={\left|(\sum_{i=1}^{n}x_{i}^{2}{)}^{2}-(\sum_{i=1}^{n}x_{i}{)}^{2}\right|}^{1/2}+\left(0.5\sum_{i=1}^{n}x_{i}^{2}+\sum_{i=1}^{n}x_{i}\right)/n+0.5$	$x_{i}\in [-100,100]$
31	$\begin{array}{l} y_{i}=\sqrt{x_{i}^{2}+x_{i+1}^{2}}\\ f_{31}(x)={\left[\frac{1}{n-1}\sum_{i=1}^{n-1}{\left(\sqrt{y_{i}}(\sin(50y_{i}^{0.2})+1)\right)}^{2}\right]}^{2} \end{array}$	$x_{i}\in [-100,100]$

**Table 18 entropy-24-01826-t018:** Test results for *f*_16_(*x*).

D	Algorithm	Worst	Best	Mean	SD
10	GSA	**927.9558**	**0.1349**	**131.5188**	**208.5184**
	GSAGJ	2.0404 × 10^3^	0.0028	416.9901	526.4530
	SGSA	4.5419 × 10^3^	0.0048	878.3686	1.1364 × 10^3^
	MACGSA	2.3696 × 10^−29^	1.4460 × 10^−34^	2.3722 × 10^−30^	5.3549 × 10^−30^
	RGLGSA	**6.4516 × 10^−45^**	**3.8488 × 10^−50^**	**9.3471 × 10^−46^**	**1.8916 × 10^−45^**
30	GSA	**318.6005**	0.0013	74.1322	80.4764
	GSAGJ	**906.5667**	1.2895	209.9172	243.1416
	SGSA	**6.8746 × 10^3^**	1.0985	1.5708 × 10^3^	1.5861 × 10^3^
	MACGSA	**8.2065 × 10^−28^**	6.1559 × 10^−33^	5.0553 × 10^−29^	1.5421 × 10^−28^
	RGLGSA	**2.8601 × 10^−42^**	**4.7112 × 10^−48^**	**1.6732 × 10^−43^**	**5.4578 × 10^−43^**
50	GSA	355.4676	0.7546	81.2026	86.5312
	GSAGJ	1.7790 × 10^3^	0.0090	305.5190	394.2691
	SGSA	5.8571 × 10^3^	0.0504	988.6499	1.4382 × 10^3^
	MACGSA	2.8593 × 10^−27^	9.2657 × 10^−31^	2.6063 × 10^−28^	5.9198 × 10^−28^
	RGLGSA	**1.0848 × 10^−42^**	**2.8539 × 10^−47^**	**1.1947 × 10^−43^**	**2.4321 × 10^−43^**

**Table 19 entropy-24-01826-t019:** Test results for *f*_17_(*x*).

D	Algorithm	Worst	Best	Mean	SD
10	GSA	8.6678 × 10^−11^	1.1821 × 10^−14^	1.3260 × 10^−11^	2.1751 × 10^−11^
	GSAGJ	9.4942 × 10^−13^	1.0772 × 10^−15^	1.7974 × 10^−13^	2.7312 × 10^−13^
	SGSA	5.5934 × 10^−8^	9.2999 × 10^−14^	2.1619 × 10^−9^	1.0164 × 10^−8^
	MACGSA	2.3449 × 10^−79^	1.8483 × 10^−93^	1.7312 × 10^−80^	5.4499 × 10^−80^
	RGLGSA	**8.8740 × 10^−108^**	**4.0256 × 10^−127^**	**4.6188 × 10^−109^**	**1.8277 × 10^−108^**
30	GSA	2.7721 × 10^9^	22.8367	1.1527 × 10^8^	5.0804 × 10^8^
	GSAGJ	6.9577 × 10^5^	59.9353	9.5043 × 10^4^	1.9611 × 10^5^
	SGSA	2.7279 × 10^8^	115.3677	1.4384 × 10^7^	5.1481 × 10^7^
	MACGSA	8.5890 × 10^−102^	1.7331 × 10^−118^	3.1893 × 10^−103^	1.5710 × 10^−102^
	RGLGSA	**5.9700 × 10^−122^**	**7.6982 × 10_−135_**	**2.0933 × 10^−123^**	**1.0888 × 10^−122^**
50	GSA	3.4420 × 10^29^	1.0669 × 10^15^	2.2045 × 10^28^	8.1793 × 10^28^
	GSAGJ	2.5940 × 10^23^	8.6661 × 10^10^	1.2069 × 10^22^	4.8832 × 10^22^
	SGSA	4.1466 × 10^24^	8.2921 × 10^13^	1.5953 × 10^23^	7.6141 × 10^23^
	MACGSA	1.7681 × 10^−107^	4.2408 × 10^−121^	6.4581 × 10^−109^	3.2318 × 10^−108^
	RGLGSA	**1.0401 × 10^−123^**	**1.5384 × 10^−144^**	**3.5856 × 10^−125^**	**1.8973 × 10^−124^**

**Table 20 entropy-24-01826-t020:** Test results for *f*_18_(*x*).

D	Algorithm	Worst	Best	Mean	SD
10	GSA	3.4437 × 10^−18^	3.1320 × 10^−19^	2.0191 × 10^−18^	7.2080 × 10^−19^
	GSAGJ	7.4174 × 10^−17^	7.2964 × 10^−18^	3.7707 × 10^−17^	1.4441 × 10^−17^
	SGSA	5.5986 × 10^−34^	3.5185 × 10^−35^	2.7418 × 10^−34^	1.1874 × 10^−34^
	MACGSA	1.5097 × 10^−33^	6.4933 × 10^−38^	1.3977 × 10^−34^	2.9910 × 10^−34^
	RGLGSA	**4.2237 × 10^−48^**	**1.4052 × 10^−51^**	**5.3557 × 10^−49^**	**9.8273 × 10^−49^**
30	GSA	22.1013	1.1135 × 10^−17^	0.7773	4.0327
	GSAGJ	93.9609	2.2794 × 10^−16^	5.1395	18.3694
	SGSA	4.0050 × 10^3^	736.0369	2.3252 × 10^3^	872.4120
	MACGSA	9.4929 × 10^−33^	3.0631 × 10^−35^	1.8175 × 10^−33^	2.6357 × 10^−33^
	RGLGSA	**2.3447 × 10^−47^**	**3.6562 × 10^−50^**	**3.2662 × 10^−48^**	**5.8753 × 10^−48^**
50	GSA	812.8504	56.8831	373.8760	184.1014
	GSAGJ	1.3323 × 10^3^	134.8774	501.4951	327.9025
	SGSA	7.1515 × 10^3^	2.8452 × 10^3^	4.9205 × 10^3^	917.0428
	MACGSA	1.5722 × 10^−32^	5.1156 × 10^−36^	1.4101 × 10^−33^	3.0031 × 10^−33^
	RGLGSA	**2.1532 × 10^−46^**	**8.1105 × 10^−50^**	**2.3062 × 10^−47^**	**5.3073 × 10^−47^**

**Table 21 entropy-24-01826-t021:** Test results for *f*_19_(*x*).

D	Algorithm	Worst	Best	Mean	SD
10	GSA	**5.6309**	5.0609	5.4357	**0.1358**
	GSAGJ	5.7058	**5.0384**	**5.3939**	0.1572
	SGSA	6.6388	6.0138	6.3808	0.1723
	MACGSA	7.0299	6.2602	6.5791	0.1821
	RGLGSA	7.4642	6.4863	6.9113	0.2123
30	GSA	26.6071	**25.5609**	26.0710	0.2053
	GSAGJ	**26.4659**	25.6065	**26.0522**	**0.1975**
	SGSA	28.2171	26.4256	26.9333	0.4014
	MACGSA	27.8766	25.8729	27.1567	0.3542
	RGLGSA	27.9882	27.0381	27.3609	0.2197
50	GSA	49.2016	45.7221	46.5439	0.6840
	GSAGJ	49.2007	**45.4268**	**46.4512**	0.7858
	SGSA	99.6238	45.6100	49.4379	10.0356
	MACGSA	48.9909	46.1487	47.8639	0.8066
	RGLGSA	**48.9011**	46.5779	47.7200	**0.5182**

**Table 22 entropy-24-01826-t022:** Test results for *f*_20_(*x*).

D	Algorithm	Worst	Best	Mean	SD
10	GSA	5.9698	**0.9950**	2.9517	**1.2389**
	GSAGJ	7.9597	**0**	3.6813	1.8318
	SGSA	9.9496	1.9899	5.5718	2.0507
	MACGSA	**0**	**0**	**0**	**0**
	RGLGSA	**0**	**0**	**0**	**0**
30	GSA	23.8790	7.9597	14.9907	3.6105
	GSAGJ	32.8336	11.9395	18.4067	4.0119
	SGSA	36.8135	10.9445	25.5041	6.6435
	MACGSA	**0**	**0**	**0**	**0**
	RGLGSA	**0**	**0**	**0**	**0**
50	GSA	42.7832	20.8941	31.9050	6.0093
	GSAGJ	57.7075	18.9042	35.0889	8.5387
	SGSA	61.6874	31.8387	47.1610	7.9322
	MACGSA	**0**	**0**	**0**	**0**
	RGLGSA	**0**	**0**	**0**	**0**

**Table 23 entropy-24-01826-t023:** Test results for *f*_21_(*x*).

D	Algorithm	Worst	Best	Mean	SD
10	GSA	2.7304	0.9393	1.5832	0.4171
	GSAGJ	3.3976	1.7065	2.6364	0.4612
	SGSA	3.1766	1.4350	2.8282	0.3456
	MACGSA	**0**	**0**	**0**	**0**
	RGLGSA	**0**	**0**	**0**	**0**
30	GSA	5.1985	2.9800	4.1999	0.6364
	GSAGJ	9.7717	5.3134	7.6816	0.9245
	SGSA	11.9302	9.2954	10.8964	0.6814
	MACGSA	**0**	**0**	**0**	**0**
	RGLGSA	**0**	**0**	**0**	**0**
50	GSA	8.3152	4.4391	6.4113	0.9250
	GSAGJ	13.0620	8.2424	10.8121	1.2371
	SGSA	20.6838	15.8774	18.6364	1.2354
	MACGSA	**0**	**0**	**0**	**0**
	RGLGSA	**0**	**0**	**0**	**0**

**Table 24 entropy-24-01826-t024:** Test results for *f*_22_(*x*).

D	Algorithm	Worst	Best	Mean	SD
10	GSA	7.3448	2	4.4654	1.5347
	GSAGJ	8	2	5.1476	1.6727
	SGSA	11.0040	3	6.7614	2.1870
	MACGSA	**0**	**0**	**0**	**0**
	RGLGSA	**0**	**0**	**0**	**0**
30	GSA	55	12	22.4959	8.1175
	GSAGJ	39	17	26.8000	5.0950
	SGSA	57	20	35.0667	8.0982
	MACGSA	141.0086	**0**	7.7670	30.2505
	RGLGSA	**0**	**0**	**0**	**0**
50	GSA	175	57	108.8667	29.6133
	GSAGJ	94	42	59.8333	12.4957
	SGSA	107	47	79.9667	14.1262
	MACGSA	850.2239	**0**	145.2359	299.2123
	RGLGSA	**0**	**0**	**0**	**0**

**Table 25 entropy-24-01826-t025:** Test results for *f*_23_(*x*).

D	Algorithm	Worst	Best	Mean	SD
10	GSA	1.0829 × 10^−18^	1.9660 × 10^−19^	5.6692 × 10^−19^	2.3275 × 10^−19^
	GSAGJ	2.7991 × 10^−17^	5.0265 × 10^−18^	1.1209 × 10^−17^	5.3271 × 10^−18^
	SGSA	**1.4998 × 10^−32^**	**1.4998 × 10^−32^**	**1.4998 × 10^−32^**	**1.1135 × 10^−47^**
	MACGSA	5.0085 × 10^−7^	1.0793 × 10^−7^	3.0263 × 10^−7^	1.0103 × 10^−7^
	RGLGSA	8.4957 × 10^−6^	1.3167 × 10^−6^	4.2384 × 10^−6^	1.6172 × 10^−6^
30	GSA	45.0420	5.2599 × 10^−18^	4.2947	10.5661
	GSAGJ	2.8179	8.8351 × 10^−17^	0.1818	0.5973
	SGSA	**0.4543**	**1.4998 × 10^−32^**	0.0665	0.1563
	MACGSA	3.2595	2.9752	3.2215	**0.0676**
	RGLGSA	0.4546	2.4474 × 10^−4^	**0.0185**	0.0840
50	GSA	177.5581	3.9951	68.4516	41.2278
	GSAGJ	18.0904	3.6758 × 10^−16^	1.3904	3.4206
	SGSA	7.7252	**4.1341 × 10^−31^**	1.4550	1.7429
	MACGSA	5.0764	4.7496	5.0298	**0.0885**
	RGLGSA	**3.2825**	0.0024	**0.6285**	0.8729

**Table 26 entropy-24-01826-t026:** Test results for *f*_24_(*x*).

D	Algorithm	Worst	Best	Mean	SD
10	GSA	1.5824 × 10^5^	65.0845	2.1881 × 10^4^	2.9856 × 10^4^
	GSAGJ	1.5669 × 10^5^	2.2076 × 10^3^	3.5142 × 10^4^	3.8005 × 10^4^
	SGSA	2.1238 × 10^5^	4.9374 × 10^3^	7.4512 × 10^4^	5.7663 × 10^4^
	MACGSA	1.4877 × 10^−32^	6.5895 × 10^−37^	1.6068 × 10^−33^	3.6474 × 10^−33^
	RGLGSA	**1.2592 × 10^−46^**	**3.9071 × 10^−51^**	**9.0954 × 10^−48^**	**2.6328 × 10^−47^**
30	GSA	2.2231 × 10^5^	1.3209 × 10^4^	7.7308 × 10^4^	5.3591 × 10^4^
	GSAGJ	2.4017 × 10_5_	9.2003 × 10^3^	8.5470 × 10^4^	6.0342 × 10^4^
	SGSA	3.1540 × 10^5^	4.3732 × 10^4^	1.4493 × 10^5^	6.9453 × 10^4^
	MACGSA	3.6955 × 10^−32^	1.8249 × 10^−35^	3.2962 × 10^−33^	7.3649 × 10^−33^
	RGLGSA	**1.1867 × 10^−46^**	6.5780 × 10^−50^	**1.3542 × 10^−47^**	**2.7548 × 10^−47^**
50	GSA	2.2231 × 10^5^	1.3209 × 10^4^	7.7308 × 10^4^	5.3591 × 10^4^
	GSAGJ	2.4017 × 10^5^	9.2003 × 10^3^	8.5470 × 10^4^	6.0342 × 10^4^
	SGSA	3.1540 × 105	4.3732 × 10^4^	1.4493 × 10^5^	6.9453 × 10^4^
	MACGSA	3.6955 × 10^−32^	1.8249 × 10^−35^	3.2962 × 10^−33^	7.3649 × 10^−33^
	RGLGSA	**1.1867 × 10^−46^**	**6.5780 × 10^−50^**	**1.3542 × 10^−47^**	**2.7548 × 10^−47^**

**Table 27 entropy-24-01826-t027:** Test results for *f*_25_(*x*).

D	Algorithm	Worst	Best	Mean	SD
10	GSA	1.6166 × 10^3^	114.4355	717.1659	407.7005
	GSAGJ	3.2975 × 10^3^	424.6107	1.6499 × 10^3^	749.6641
	SGSA	3.0122 × 10^3^	291.2547	1.5685 × 10^3^	670.7036
	MACGSA	5.3094 × 10^−33^	2.1696 × 10^−37^	2.4561 × 10^−34^	9.6672 × 10^−34^
	RGLGSA	**7.6013 × 10^−48^**	**1.4909 × 10^−51^**	**6.8777 × 10^−49^**	**1.7501 × 10^−48^**
30	GSA	2.2537 × 10^3^	299.4626	872.2012	393.5168
	GSAGJ	2.3856 × 10^3^	3.3096 × 10^−16^	486.5437	610.2161
	SGSA	3.2041 × 10^−32^	5.0516 × 10^−33^	1.3777 × 10^−32^	6.3954 × 10^−33^
	MACGSA	3.6098 × 10^−32^	5.0320 × 10^−36^	2.0449 × 10^−33^	6.5866 × 10^−33^
	RGLGSA	**2.1523 × 10^−47^**	**4.3108 × 10^−50^**	**3.0550 × 10^−48^**	**5.7203 × 10^−48^**
50	GSA	2.8219 × 10^3^	566.1510	1.7062 × 10^3^	645.7675
	GSAGJ	2.6789 × 10^3^	1.3496 × 10^−15^	722.0980	788.4850
	SGSA	2.0515 × 10^−30^	8.0778 × 10^−32^	4.2586 × 10^−31^	4.3106 × 10^−31^
	MACGSA	1.3764 × 10^−32^	4.2468 × 10^−36^	1.3404 × 10^−33^	3.3730 × 10^−33^
	RGLGSA	**6.3641 × 10^−47^**	**1.1221 × 10^−49^**	**6.7762 × 10^−48^**	**1.5149 × 10^−47^**

**Table 28 entropy-24-01826-t028:** Test results for *f*_26_(*x*).

D	Algorithm	Worst	Best	Mean	SD
10	GSA	2.7037 × 10^−9^	1.3481 × 10^−9^	1.8639 × 10^−9^	3.3343 × 10^−10^
	GSAGJ	9.7995 × 10^−9^	4.6786 × 10^−9^	7.5941 × 10^−9^	1.3530 × 10^−9^
	SGSA	7.9936 × 10^−15^	4.4409 × 10^−15^	4.5593 × 10^−15^	6.4863 × 10^−16^
	MACGSA	**8.8818 × 10^−16^**	**8.8818 × 10^−16^**	**8.8818 × 10^−16^**	**0**
	RGLGSA	**8.8818 × 10^−16^**	**8.8818 × 10^−16^**	**8.8818 × 10^−16^**	**0**
30	GSA	4.5600 × 10^−9^	2.8379 × 10^−9^	3.5839 × 10^−9^	4.5843 × 10^−10^
	GSAGJ	1.7140 × 10^−8^	1.0784 × 10^−8^	1.4658 × 10^−8^	1.6459 × 10^−9^
	SGSA	2.2204 × 10^−14^	7.9936 × 10^−15^	1.4744 × 10^−14^	3.5343 × 10^−15^
	MACGSA	**8.8818 × 10^−16^**	**8.8818 × 10^−16^**	**8.8818 × 10^−16^**	**0**
	RGLGSA	**8.8818 × 10^−16^**	**8.8818 × 10^−16^**	**8.8818 × 10^−16^**	**0**
50	GSA	8.2910 × 10^−9^	3.6928 × 10^−9^	4.9393 × 10^−9^	8.8854 × 10^−10^
	GSAGJ	2.4806 × 10^−8^	1.5528 × 10^−8^	1.9258 × 10^−8^	2.2218 × 10^−9^
	SGSA	5.7732 × 10^−14^	1.5099 × 10^−14^	2.9073 × 10^−14^	7.5758 × 10^−15^
	MACGSA	**8.8818 × 10^−16^**	**8.8818 × 10^−16^**	**8.8818 × 10^−16^**	**0**
	RGLGSA	**8.8818 × 10^−16^**	**8.8818 × 10^−16^**	**8.8818 × 10^−16^**	**0**

**Table 29 entropy-24-01826-t029:** Test results for *f*_27_(*x*).

D	Algorithm	Worst	Best	Mean	SD
10	GSA	0.0762	**0**	0.0200	0.0205
	GSAGJ	0.0270	**0**	0.0030	0.0061
	SGSA	0.0172	**0**	0.0031	0.0056
	MACGSA	**0**	**0**	**0**	**0**
	RGLGSA	**0**	**0**	**0**	**0**
30	GSA	9.5227	1.7763	4.2129	1.7361
	GSAGJ	0.0515	**0**	0.0038	0.0108
	SGSA	0.0148	**0**	0.0015	0.0040
	MACGSA	**0**	**0**	**0**	**0**
	RGLGSA	**0**	**0**	**0**	**0**
50	GSA	25.1669	11.4594	17.4486	4.1201
	GSAGJ	0.1761	**0**	0.0356	0.0477
	SGSA	0.0148	**0**	4.9241e-04	0.0027
	MACGSA	**0**	**0**	**0**	**0**
	RGLGSA	**0**	**0**	**0**	**0**

**Table 30 entropy-24-01826-t030:** Test results for *f*_28_(*x*).

D	Algorithm	Worst	Best	Mean	SD
10	GSA	1.1961 × 10^−10^	6.7969 × 10^−11^	1.0001 × 10^−10^	1.1193 × 10^−11^
	GSAGJ	1.2086 × 10^−10^	6.9683 × 10^−11^	9.8470 × 10^−11^	1.2157 × 10^−11^
	SGSA	**0**	**0**	**0**	**0**
	MACGSA	**0**	**0**	**0**	**0**
	RGLGSA	**0**	**0**	**0**	**0**
30	GSA	5.8032 × 10^−11^	4.6224 × 10^−11^	5.2202 × 10^−11^	3.1700 × 10^−12^
	GSAGJ	5.6719 × 10^−11^	4.7120 × 10^−11^	5.2218 × 10^−11^	2.6389 × 10^−12^
	SGSA	5.8824 × 10^−13^	**0**	1.2443 × 10^−13^	1.9250 × 10^−13^
	MACGSA	**0**	**0**	**0**	**0**
	RGLGSA	**0**	**0**	**0**	**0**
50	GSA	3.5579 × 10^−11^	3.1050 × 10^−11^	3.3152 × 10^−11^	9.9662 × 10^−13^
	GSAGJ	3.4840 × 10^−11^	2.7680 × 10^−11^	3.3151 × 10^−11^	1.4994 × 10^−12^
	SGSA	1.2293 × 10^−12^	8.3267 × 10^−15^	4.3324 × 10^−13^	3.1526 × 10^−13^
	MACGSA	**0**	**0**	**0**	**0**
	RGLGSA	**0**	**0**	**0**	**0**

**Table 31 entropy-24-01826-t031:** Test results for *f*_29_(*x*).

D	Algorithm	Worst	Best	Mean	SD
10	GSA	0.5050	0.1770	0.3001	0.0798
	GSAGJ	**0.3542**	**0.1485**	**0.2515**	**0.0563**
	SGSA	0.5103	0.2174	0.3332	0.0810
	MACGSA	1.5100	0.4278	0.7252	0.2975
	RGLGSA	0.6482	0.3033	0.4403	0.0917
30	GSA	**0.1652**	**0.0440**	**0.1174**	**0.0290**
	GSAGJ	0.1938	0.0815	0.1362	0.0292
	SGSA	0.2739	0.1108	0.1815	0.0425
	MACGSA	0.4296	0.1637	0.2719	0.0714
	RGLGSA	0.3036	0.1204	0.2153	0.0409
50	GSA	0.5050	0.1770	0.3001	0.0798
	GSAGJ	**0.3542**	**0.1485**	**0.2515**	**0.0563**
	SGSA	0.5103	0.2174	0.3332	0.0810
	MACGSA	1.5100	0.4278	0.7252	0.2975
	RGLGSA	0.6482	0.3033	0.4403	0.0917

**Table 32 entropy-24-01826-t032:** Test results for *f*_30_(*x*).

D	Algorithm	Worst	Best	Mean	SD
10	GSA	0.5008	0.3570	**0.4925**	0.0267
	GSAGJ	0.5010	0.4387	0.4944	0.0175
	SGSA	0.5009	**0.3110**	0.4895	0.0412
	MACGSA	**0.5000**	0.3965	0.4860	0.0285
	RGLGSA	**0.5000**	0.4574	0.4981	**0.0081**
30	GSA	0.5013	0.4344	0.4884	**0.0159**
	GSAGJ	0.5010	0.3233	0.4581	0.0495
	SGSA	0.5016	**0.2767**	**0.4204**	0.0553
	MACGSA	**0.5000**	0.3568	0.4661	0.0438
	RGLGSA	**0.5000**	0.4159	0.4851	0.0263
50	GSA	0.6156	**0.2404**	0.3872	0.0743
	GSAGJ	**0.4495**	0.2743	**0.3814**	0.0405
	SGSA	0.5687	0.2730	0.3726	0.0542
	MACGSA	0.5000	0.3422	0.4711	0.0465
	RGLGSA	0.5000	0.4451	0.4900	**0.0154**

**Table 33 entropy-24-01826-t033:** Test results for *f*_31_(*x*).

D	Algorithm	Worst	Best	Mean	SD
10	GSA	1.0420 × 10^−4^	2.7999 × 10^−34^	4.6894 × 10^−6^	1.9938 × 10^−5^
	GSAGJ	2.8974 × 10^−7^	2.1420 × 10^−32^	1.9555 × 10^−8^	7.1460 × 10^−8^
	SGSA	1.9305 × 10^−5^	**0**	7.0894 × 10^−7^	3.5154 × 10^−6^
	MACGSA	1.1429 × 10^−21^	5.5902 × 10^−24^	4.5938 × 10^−22^	3.0559 × 10^−22^
	RGLGSA	**7.1484 × 10^−30^**	2.1262 × 10^−32^	**1.0346 × 10^−30^**	**1.3575 × 10^−30^**
30	GSA	0.0786	1.5937 × 10^−5^	0.0118	0.0185
	GSAGJ	5.8248 × 10^−4^	1.3017 × 10^−10^	7.2445 × 10^−5^	1.4989 × 10^−4^
	SGSA	0.0261	1.4485 × 10^−6^	0.0040	0.0063
	MACGSA	7.9223 × 10+	1.4338 × 10^−23^	2.2198 × 10^−22^	1.9444 × 10^−22^
	RGLGSA	**1.1549 × 10^−30^**	**9.2467 × 10^−33^**	**3.4346 × 10^−31^**	**2.8391 × 10^−31^**
50	GSA	0.0785	6.1868 × 10^−4^	0.0159	0.0207
	GSAGJ	0.0189	1.0070 × 10^−5^	0.0031	0.0050
	SGSA	0.0548	5.2674 × 10^−4^	0.0101	0.0146
	MACGSA	3.8503 × 10^−19^	9.9380 × 10^−21^	1.2760 × 10^−19^	8.0890 × 10^−20^
	RGLGSA	**1.0666 × 10^−25^**	**6.5193 × 10^−28^**	**1.6474 × 10^−26^**	**2.0978 × 10^−26^**

## Data Availability

Data is available upon reasonable request from the author.

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
