# Peer review of "Improved Gravitational Search Algorithm Based on Adaptive Strategies"

_entropy, 2022, doi:10.3390/e24121826_

Round 1
Reviewer 1 Report
· The authors proposed an improved version of Gravitational Search Algorithm (GSA). Furthermore, the paper lacks convincing simulated analyses which demonstrate superior performance over prior art.
· The authors should compare it against known art of Gravitational Search Algorithm, on the basis of typical performance of modified GSA’s.
· Not cited Ref [18]. Zhenkai, X. Modification and Application of Gravitational Search Algorithm. Master’s; University of Shanghai for Science & Technology, 2014.
Author Response
Point 1: The authors proposed an improved version of Gravitational Search Algorithm (GSA). Furthermore, the paper lacks convincing simulated analyses which demonstrate superior performance over prior art.
Response 1: This problem is a defect in this paper, and also a prospect for future research. No Free Lunch Theory is a very important theorem in the field of optimization research, which reflects that no optimization algorithm can outperform other algorithms in average performance on all optimization problems. The improved algorithm in this paper also has this problem, and the experimental analysis cannot fully cover many optimization problems, which is convincing. The future research direction is mainly to optimize the overall performance of the algorithm, so that the performance of the improved algorithm is better than that of other algorithms on as many optimization problems as possible.
Point 2: The authors should compare it against known art of Gravitational Search Algorithm, on the basis of typical performance of modified GSA’s.
Response 2: The experiments in Section 5.1 show that the improved algorithm maintains a high performance advantage in the optimization of unimodal and multimodal functions. In Section 5.2, benchmark test functions are selected to compare and test the algorithm, and compared with the basic gravity search algorithm and the current highly efficient improved gravity search algorithm, further proving that the algorithm has greatly improved in the accuracy, stability and convergence of optimization.
Point 3: Not cited Ref [18]. Zhenkai, X. Modification and Application of Gravitational Search Algorithm. Master’s; University of Shanghai for Science & Technology, 2014.
Response 3: Reference number has been modified, see the manuscript.

Reviewer 2 Report
The authors provide sufficient background knowledge and literature reviews of this paper. The writing is clear and reasonable. However, there are some technical problems, mainly on the careless uses of notations. For example,
ž P3L128: The last superscript should be $D$, not $N$.
ž P4L165: The term under the fraction (i.e. the denominator) should be $M_i(t)$, not $M_ij(t)$.
ž P5L205, L208: Please use $N$ instead of $n$ for consistency.
P7L320: Actually, O(N(N-1)+9N) can be simplified as O(N^2).
Another problems are related to the description of the contents. For example,
ž P5L205: This definition is quite weird. The formula defines “the average distance” $\delta$, but the given name is “population density”. The meaning of these two terms are actually reciprocal concept in physics. The misused term would cause the reading misleading.
In Tables 2 to 6 and Table 12, there is a problem in the column “Best”. An entry is missing in every column in which the numbers listed are not aligned properly.
Finally, I think the proof of Theorem 4.1 is rigorous. How can you make sure that X_n will approach the global optimal solution X^* within $\epsilon$? BTW, the formula in L344 should be $X_{k+1}=f(X_k)$. The use of notation “there exists” before $\rho$ is not proper.
Author Response
Point 1: The authors provide sufficient background knowledge and literature reviews of this paper. The writing is clear and reasonable. However, there are some technical problems, mainly on the careless uses of notations. For example,
P3L128: The last superscript should be $D$, not $N$.
P4L165: The term under the fraction (i.e. the denominator) should be $M_i(t)$, not $M_ij(t)$.
P5L205, L208: Please use $N$ instead of $n$ for consistency.
P7L320: Actually, O(N(N-1)+9N) can be simplified as O(N^2).
Response 1: Modified, see the uploaded manuscript for details.
Point 2: P5L205: This definition is quite weird. The formula defines “the average distance” $\delta$, but the given name is “population density”. The meaning of these two terms are actually reciprocal concept in physics. The misused term would cause the reading misleading.
Response 2: The definition of population density is the median of the average distance of all particles; The average distance of particles is defined as the average value of particles and other particles.
Point 3: In Tables 2 to 6 and Table 12, there is a problem in the column “Best”. An entry is missing in every column in which the numbers listed are not aligned properly.
Response 3: Modified, see the uploaded manuscript for details.
Point 4: Finally, I think the proof of Theorem 4.1 is rigorous. How can you make sure that X_n will approach the global optimal solution X^* within $\epsilon$? BTW, the formula in L344 should be $X_{k+1}=f(X_k)$. The use of notation “there exists” before $\rho$ is not proper.
Response 4: The essence of algorithm optimization is an iterative process. Under the action of gravity, the individuals in the algorithm attract each other, forcing small mass individuals to constantly move to larger mass individuals to find the optimal solution X. At this time, X is not necessarily the global optimal solution. Modified,see the uploaded manuscript for details.
